# Assessment of Flood Hazard in Climatic Extreme Considering Fluvio-Morphic Responses of the Contributing River: Indications from the Brahmaputra-Jamuna's Braided-Plain

**Shampa** [1,*], **Binata Roy** [1,2], **Md. Manjurul Hussain** [1], **A. K. M. Saiful Islam** [1], **Md. Ashiqur Rahman** [3] and **Khaled Mohammed** [1]

1    Institute of Water and Flood Management (IWFM), Bangladesh University of Engineering and Technology (BUET), Dhaka 1000, Bangladesh
2    Civil Engineering, University of Virginia, Charlottesville, VA 22904, USA
3    Department of Civil and Environmental Engineering, Florida International University, Miami, FL 33199, USA
*    Correspondence: shampa_iwfm@iwfm.buet.ac.bd

**Abstract:** Climate change is expected to raise river discharge and sea level in the future, and these near-term changes could alter the river flow regime and sedimentation pattern of future floods. Present hazard assessment studies have limitations in considering such morpho-dynamic responses in evaluating flood hazards or risks. Here, we present a multi-model-based approach to quantify such potential hazard parameters influenced by climate change for the most vulnerable communities living on river bars and islands of the Brahmaputra–Jamuna River. River flood-flow and flood wave propagation characteristics are predicted to be affected by changing temporal distribution patterns of precipitation as a result of enhanced global warming. Increased incidences of large multi-peak floods or uncommon floods resulting in long-duration floods driven by sea-level rise may happen as a result of this. To assess it, we have set up a hydromorphic model, Delft3D, for the Brahmaputra–Jamuna River forced by upstream flow, generated from a hydrological model SWAT, over the Brahmaputra basin. The simulations cover moderate, wettest, and driest conditions of the RCP8.5 scenario, and the results reflect the flooding consequences of the near-future, mid-century, and end-century. Floods in the Brahmaputra–Jamuna River are becoming more severe, frequent, and long-lasting, as a result of climate change, and are expected to last until the end of November rather than the current September timeline. While assessing the hazard, we found that the pattern and timing of the flood are as equally important as the peak of the flood, as the river continuously adjusts its cross-sectional area with the flow. The study also demonstrates that, depending on their location/position, climate-induced hazards can affect sand bars/islands disproportionally. The high flood depth, duration, and sedimentation have a significant impact on the sand bars downstream of the river, making them more vulnerable.

**Keywords:** RCP 8.5; climate-change; Brahmaputra-Jamuna; Delft3D; flood hazard; SWAT

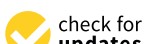



## 1. Introduction

Extreme weather events are likely to become more frequent and larger as a result of climate change [1]. Therefore, large-scale river flooding or events such as pluvial flooding will be more frequent. The hydrology and morphology of river systems are predicted to be affected by changes in precipitation levels, glacial mass balance, and the extent of permafrost [2]. As a result, the amount, timing, distribution, and geographic location of large-scale river floods are becoming highly unpredictable [3]. Classifying the historical flood events as regular or extreme can be a matter of discussion, as the flood is a location-specific phenomenon [4,5]. For example, communities living adjacent to large rivers (i.e., deltaic regions) may be used to high flood depths with low flood velocity. On the other hand, for the communities living far away from the river, a similar flood event can be more

hazardous [6,7]. Furthermore, due to ongoing socio-economic developments, the impact of floods in the coming decades could substantially increase [8,9]. Therefore, the flood risk management (FRM) must deal with these utilities; the present flood hazard assessment should be combined with the expected flood consequences, which will continue to change due to the alteration in flood hydro-meteorological drivers, e.g., temperature/rainfall, land-use patterns, and socio-economic development [10]. In line with this, Flood hazard maps and flood risk assessment maps can be effective tools for FRM in preparedness, contingency, and recovery stages [11].

Now the vital question arises: is the current 'Flood hazard assessment' (the estimation of overall adverse effects of flooding for a particular area) considered among the hazard elements that would be altered due to climate change? As an example, we have summarized the element of flood hazards taken care of in the past studies in Bangladesh, which is one of the most vulnerable countries of the flood disaster. In general, the flood hazard parameters include one or more components of the depth of flooding, duration of flooding, flood wave velocity, and rate of rising of water level, depending on the characteristics of the study area [12]. The depth of inundation was the most commonly used parameter (i.e., refs. [13–19]). Areal extent was the next commonly used parameter, in addition to the flood depth (i.e., refs. [14,17,19]). Flood duration was considered only in a few studies ([14,19]). Very few studies considered flood frequencies in addition to other parameters [15]. Most of the time, other components were overlooked in favor of flood depth due to their minor contribution to flood hazards without any quantifications of the other components such as velocity or sedimentation. Globally, such practices in other river flood studies are also found (i.e., refs. [20–22]). The geophysical location of the focused flood-affected area mentioned above may not be severely affected by the modification of river morphology or sedimentation. However, in the case of flood disasters where the flood debris is a greater concern, the modification of hazard components, including the debris parameter, is quite common (example refs. [1,23,24])

As Bangladesh is situated at the most downstream part of the Ganges–Brahmaputra–Meghna (GBM) delta, it will be affected upstream and downstream during climate-induced extreme flooding disasters: intense flooding due to high precipitation, a drainage problem due to sea-level rise, and an associated morphological change of the outfall of the river [10,25,26]. The previous studies reported that, due to the increase in global warming level from present-day to 4 °C, the major rivers of GBM are subjected to experience increased (up to 60%) fluvial flow [27,28]. Haque et al. [29] show that, due to climate change, the flow in the Brahmaputra basin may increase from 18% to 61%, while the sediment load may increase from 34% to 115%. Mohammed et al. [30] observed the higher frequency and magnitude of floods in the Brahmaputra basin due to several climate projections; not only this, but the annual minimum flow may also increase by up to 24% at the end century compared to the flow of 1980–2010 [31]. Though the increase in discharge is plausible, the future sediment load is uncertain due to anthropogenic interventions in the basin [32,33]. All these alterations will affect the chars (local Bengali name of bar/island of a river) of the river to some degree. Nonetheless, the Northern Indian Ocean, including the Bay of Bengal, is said to be experiencing a faster rate of Sea Level Rise (SLR) than other oceans, which will contribute to the severity of fluvio-tidal and storm surge-related floods in the coastal region [28,34]. Rahman et al. [35] show that, for a category 4 cyclone, the inundation area is likely to increase by 1.1% to 5.8% for the SLR of 0.5 m to 1.5, correspondingly. Furthermore, most previous climatic model-related studies have been 'region' or 'basin'-specific, with little attention paid to river morphology-specific phenomena due to spatial downscaling issues [27,30,36,37]. Against this backdrop, this research attempts to identify flood hazards in climatic extremes considering the fluvio-morphic responses of the river, Brahmaputra–Jamuna, using the multi-model approach. This goal will be accomplished, firstly, by generating basin-scale climate-induced floods for several timelines using the hydrologic model, as well as quantifying the severity of climate-induced floods. Secondly, using the hydro-morphic model, the climate-induced floods have been simulated, and from

the simulation results along with the usual hazard parameters (i.e., depth, duration, and velocity), the morpho-dynamic component of hazard (extreme sedimentation in this case) will be quantified. Finally, the climate-induced flood hazard will be assessed in a combination of hydro-morpho dynamic components with severity. As a test case area, the chars of the Brahmaputra–Jamuna River of Bangladesh were selected where the morpho-dynamic hazard impact is quite high [38,39]. The following sections describe the details of the study area, methods, results, discussion, and conclusion.

## 2. Study Area

The transboundary river Brahmaputra–Jamuna is one of the main freshwater sources of South Asia, contributing to the livelihood of around 66 million people through agriculture and potable water [27]. Originating from a Himalayan glacier in southwest Tibet at an altitude of 5100 m MSL, the mighty Brahmaputra flows almost 3200 km in China (named Yarlung) and India (named Dihang later the Brahmaputra) before entering Bangladesh [27,40]. The total catchment area is nearly 543,400 km$^2$ in Bangladesh (here named the Brahmaputra–Jamuna River), which possesses only 7% of the catchment [40]. Along its whole path, the river is braided along almost all of its course (Indian and Bangladesh part) with numerous bars and channels [36–38]. Therefore, this study concentrates on the chars of the Brahmaputra–Jamuna in the Bangladesh region. The reach length is nearly 225 km long with a population density of 0.4 to 182 per person per 80 m$^2$ [41]. These chars are extremely resourceful with agriculture and livestock mainstays [33]. Nearly 300 chars are observed during the year 2020, as shown in Figure 1a, and they are mostly compound and unit bars in nature [40]. The river discharge shows strong seasonal variabilities (Figure 1b). In the last 20 years, the occurrence of extreme floods is quite increased, indicating the influence of climate change along with other external drivers, i.e., flow alterations or dam constructions in upstream countries [42]. Hofer and Messerli [43] investigated the causes of the severe flooding in Bangladesh and found that the large-scale flooding is caused by the simultaneous peak discharges of the major rivers such as the Ganges and Brahmaputra–Jamuna, significant runoff from the Meghalaya Hills, excessive rainfall in Bangladesh, high groundwater tables, and high spring tides. They also pointed to the abolition of natural water storage due to rapid population growth in the lowlands, and the construction of lateral river embankments such as the Brahmaputra Right Embankment (BRE) appears to have a substantial effect on the flooding processes. Grumbine and Pandit [44] mentioned the planning of construction of hydroelectric dams in the Indian part of the basin, which are supposed to alter the flow-sediment balance of the river in the future. The community also responded that they are experiencing multiple high-peak floods with longer durations nowadays [45]. IWFM-NRP [45] also reported unusual sedimentation above the agricultural land during recent floods. Within reach, the braided index (total channel length vs major channel length) varies from 4.1 to 5.6, whereas the char land persistency (time length from emergence) varies from 1 to 27 years.

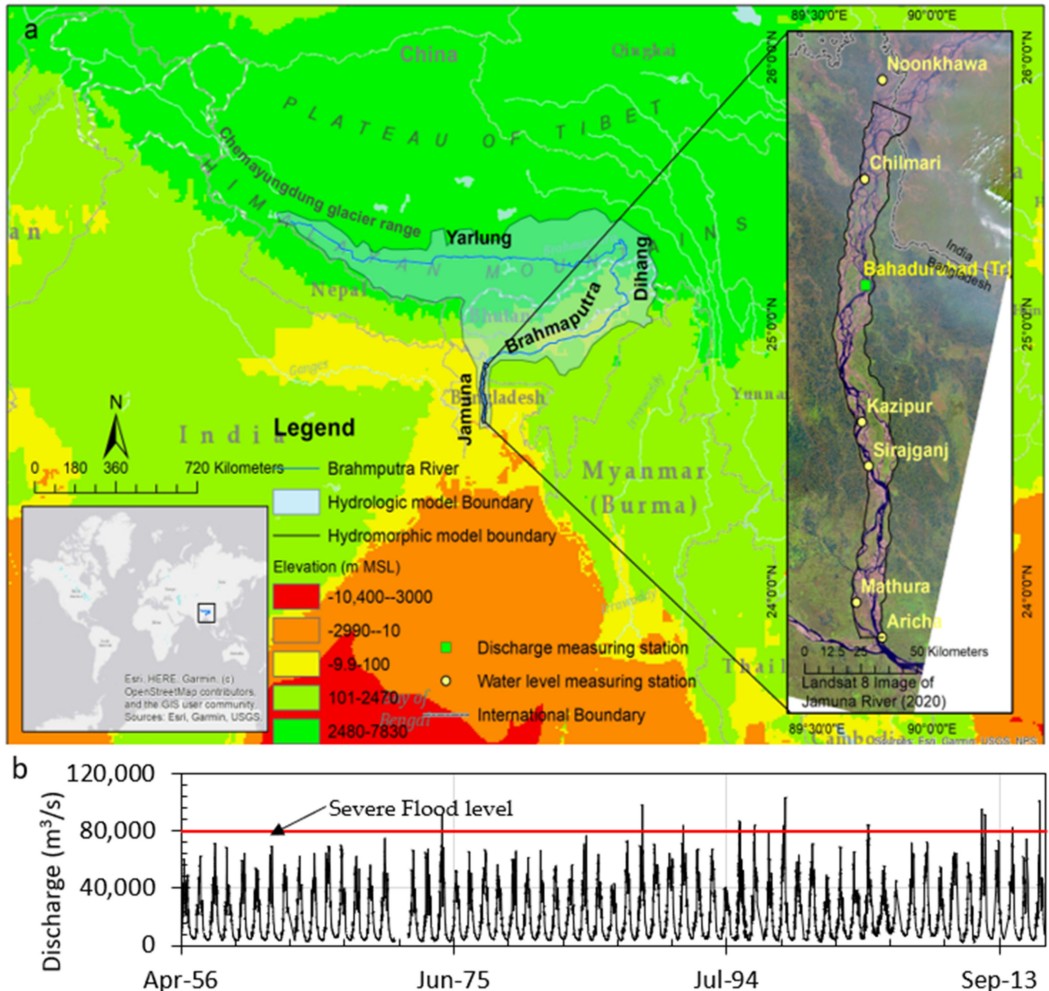

**Figure 1.** Details of the study area. (**a**) Brahmaputra–Jamuna River, along with Hydrologic and Hydromorphic model boundary areas. (**b**) Time-series Discharge hydrograph of the Brahmaputra–Jamuna River at Bahadurabad station for the last forty years.

## 3. Methods and Materials

This study links a hydrologic model with a hydromorphic model to assess the impact of climate change in terms of flood hazards. Figure 2 shows the overall methodological framework for estimating the climate, hydrology, and morphological changes for extreme climate events. Here, GCM-driven Temperature and Precipitation for the RCP 8.5 scenario were given as inputs, along with the soil and land cover data, in the hydrological model. The hydrological model discharge outputs for nine future scenarios have been generated as the moderate, driest, and wettest conditions. Instead of comparing climate scenarios with a model-generated baseline, we used the measured data as a baseline. Using the available time-series data (1956 to 2006), one hydrograph has been generated for the return period of 2.33 per year, which is called the average condition or base condition through the paper (see [46] for the methodology of generation of hydrograph for average condition). One of the focuses of our study is to compare future flooding with the flood that is commonly experienced in the study area. Moreover, the annual daily discharge hydrographs for the baseline periods for each of the ensembles (Moderate (R1i1p1), Driest (R2i1p1), and Wettest (R7i1p1)) of RCP 8.5 scenarios are comparable with the annual average daily discharge for the 2.33 per year return period (Figure S1). The corresponding water level of future scenarios has been generated using the rating curve (see Figure S2, using Kennedy [47]), which was adjusted by the sea-level change due to RCP 8.5. These discharges are used, as the upstream and water level hydrographs are used, as the downstream boundary

conditions of the model. Table 1 listed the cases and criteria considered in this study. From the simulation results, flood depth, velocity, duration, and sedimentation were calculated. Severity was derived from the time series data analysis of all scenarios.

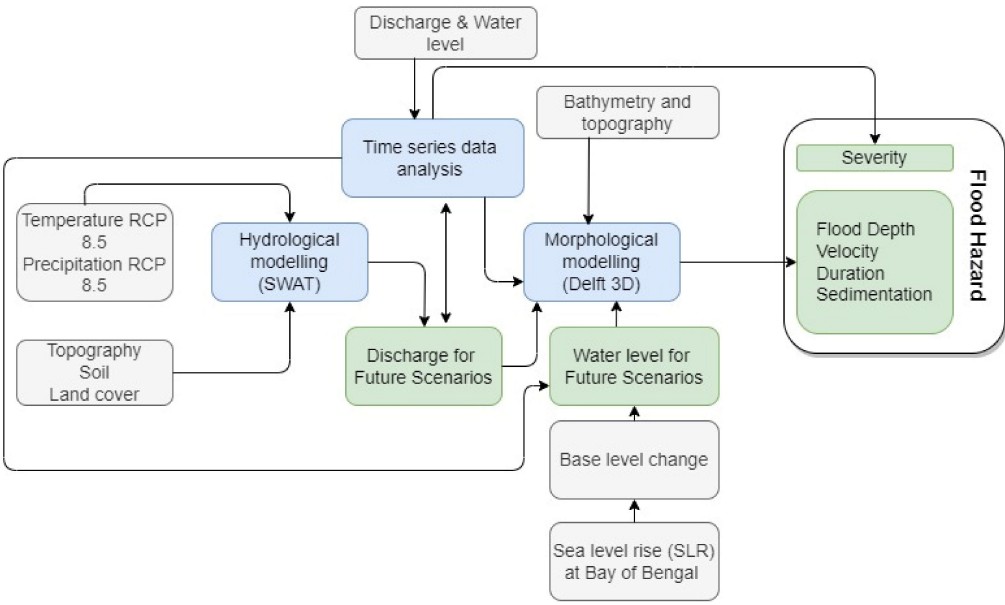

**Figure 2.** Methodological framework for assessing the climate, hydrology, and morphological changes in extreme climate events.

**Table 1.** Cases and criteria considered in this study.

| Cases | Condition | Timeline | Criteria | | | |
|---|---|---|---|---|---|---|
| | | | Flow (Upstream Boundary Condition) | Source | Water Level (WL) (Downstream Boundary Condition) | Source |
| Average condition | Base | 1956–2016 | Average flow condition (return period- 2.33 years) | Observed time series data | WL for Average flooding condition (return period- 2.33 years) | Observed time series data (IPCC 2014) |
| RCP 8.5 | Moderate | Near-future (2020s) | 90th percentile of daily flow considering years 2006–2035 | SWAT model | WL + Projected SLR where SLR = 0.17 m | IPCC AR5 report |
| | | Mid-Century (2050s) | 90th percentile of daily flow considering years 2036–2065 | | SLR = 0.38 m | |
| | | End-Century (2080s) | 90th percentile of daily flow considering years 2066–2095 | | SLR = 0.82 m | |
| | Driest | Near-future (2020s) | 90th percentile of daily flow considering years 2006–2035 | | SLR = 0.17 m | |
| | | Mid-Century (2050s) | 90th percentile of daily flow considering years 2036–2065 | | SLR = 0.38 m | |
| | | End-Century (2080s) | 90th percentile of daily flow considering years 2066–2095 | | SLR = 0.82 m | |

**Table 1.** *Cont.*

| Cases | Condition | Timeline | Criteria | | | |
|-------|-----------|----------|----------|--------|---------------|--------|
| | | | Flow (Upstream Boundary Condition) | Source | Water Level (WL) (Downstream Boundary Condition) | Source |
| | Wettest | Near-future (2020s) | 90th percentile of daily flow considering years 2006–2035 | | SLR = 0.17 m | |
| | | Mid-Century (2050s) | 90th percentile of daily flow considering years 2036–2065 | | SLR = 0.38 m | |
| | | End-Century (2080s) | 90th percentile of daily flow considering years 2066–2095 | | SLR = 0.82 m | |

## 3.1. Hydrological Simulation

### 3.1.1. Climate Model Selection

Climate modelling studies are highly dependent on the climatic projections simulated by different Global Climate Models (GCMs) and Regional Climate Models (RCMs). The details of the climate model and selection procedure can be found in Supplementary Section S1.1. This EC-EARTH3-HR GCM has seven different realizations/variants enforced with different SSTs and SICs, as tabulated in Table 2. There are seven ensembles in this study. The ensemble member is defined as 'rNiMpL', where N is the number of realizations, M is the number of different initialization states, and L is the number of used physical parameterizations. For example, R5i1p1 means the number of realization members = 5, the number of different initialization states = 1, and the number of used physical parameterizations = 1 for this ensemble. For details, kindly see Taylor [48].

**Table 2.** Selection of Configuration/Initialization for Climate Change Impact Analysis.

| GCM | Enforcing Models by SST and SIC | Ensemble Members | % Increase in 2020s (2006–2035) Compared to Baseline (1976–2005) | % Increase in 2050s (2035–2065) Compared to Base-Line (1976–2005) | % Increase in 2080s (2066–2095) Compared to Base-Line (1976–2005) | Remarks |
|-----|---------------------------------|------------------|------|------|------|---------|
| EC-EARTH3-HR | IPSL-CM5A-LR | R2i1p1 | 3.1 | 7.4 | 12.8 | Driest |
| | GFDL-ESM2M | R4i1p1 | 2.1 | 13.6 | 15.0 | |
| | HadGEM2-ES | R5i1p1 | 4.5 | 15.0 | 19.6 | |
| | EC-EARTH | R1i1p1 | 3.6 | 13.6 | 32.8 | Moderate |
| | GISS-E2-H | R3i1p1 | 6.6 | 15.9 | 35.1 | |
| | IPSL-CM5A-LR | R6i1p1 | 8.2 | 11.2 | 35.2 | |
| | HadCM3/abuig (Amazon dieback) | R7i1p1 | 8.2 | 19.9 | 37.0 | Wettest |

### 3.1.2. Selection of Specific Initializations

For future flow generation, the SWAT model has been simulated with the EC-EARTH HR daily precipitation and maximum/minimum temperature data, from 1976 to 2100, for all the ensembles under the RCP 8.5 Scenario mentioned in Table 2. Then, the daily flow hydrographs for each of the years are extracted from the SWAT model at Bahadurabad. There are four periods (each spanning thirty-year), such as the baseline (1976–2005), 2020s (2006–2035), the 2050s (2036–2065), and the 2080s (2066–2095), selected for the hydrologic model flow. Then, the percent increase in flow, for each of the time periods, has been computed with respect to the baseline, as shown in Table 2. Finally, three ensembles—r2i1p1, r1i1p1, and r7i1p1—are selected from the seven different ensembles based on driest, moderate, and wettest flow conditions.

After selecting specific ensembles, the annual daily flow hydrographs of each of the periods—the 2020s, 2050s, and 2080s—have been prepared to simulate the hydrodynamic model, which is done by taking the 90th percentile of daily flow values from each of the time spans (2020s, 2050s, and 2080s). As this study is mainly focused on the extreme flood scenario, maximum flood hydrographs should serve better than the average values. However, the maximum of the climate dataset is not recommended in climate-related studies as it contains outliers. Hence, extreme flood scenarios are defined by the 90th percentile (Roy et al., 2021) and, later, used as the flow input for the hydrodynamic model. For the 2080s of RCP 8.5, the highest discharge is found as 102,450 $m^3/s$, 98,881 $m^3/s$, and 83,464 $m^3/s$ for the wettest, moderate, and dried ensembles, respectively. For future water level generation, the sea-level-rise (SLR) at Bay of Bengal is defined as 17, 38, and 82 cm for 2020s, 2050s, and 2080s of all ensembles, respectively, from the IPCC AR5 report (IPCC 2014). Then, the adjustment of sea-level rise at the exact hydrodynamic model d/s boundary (Aricha) was made using the base level adjustment mentioned in ref. [26].

### 3.1.3. The Schematization of the Hydrologic Model

A hydrologic model of the Brahmaputra basin, developed by ref. [28] in SWAT, has been used to estimate the future flow at Bahadurabad Transit of the Brahmaputra River (Figure 1a). The model topography has been set up using HydroSHEDS 90m DEM [49]. The GlobCover land use map prepared by the European Space Agency [50] and the soil map prepared by the Food and Agricultural Organization [51] have been used as land-use and soil information for the model, respectively. Daily precipitation and temperature data from the Princeton Global Forcing (version 2) dataset [52] of the period 2001 to 2012 are used during the development of the SWAT model.

### 3.1.4. Hydrologic Model Validation

Before calibrating the model, sensitivity analyses are performed on all hydrology parameters of SWAT using SWAT-CUP. The hydrologic model has been calibrated for 2001–2006 and validated for 2007–2012, compared against the observed discharge data of the Bangladesh Water Development Board (BWDB). Please see Supplementary Section S1.1.4 for detailed validation of the model.

### *3.2. Hydromorphic Simulation*

A physics-based 2D morpho-dynamic model of the Brahmaputra–Jamuna that is well-calibrated and validated [53,54] has been used to assess the impacts of several climatic scenarios over the char. The numerical model was used on the open-source platform of Delft3D (flow version 4.00.01.000000) ([55]). Please see Supplementary Section S1.2 for the details hydromorphic model.

For the numerical model, a 225 km-long curvilinear grid was constructed with an average width of 13 km, starting from almost 10 km downstream of the water level measuring station at Noonkhawa and ending near the water level measuring station at Aricha, as shown in Figure 3. Grid cells of 1117 × 73 were used to discretize the reach. The bar sizes varied from 549 × 205 $m^2$ to 28,635 × 10,475 $m^2$ within reach of the Brahmaputra–Jamuna River, so this grid resolution was chosen to cover every bar by at least two grid cells. Orthogonal curvilinear grid was generated in Cartesian coordinates systems where the average grid cell size was 201 × 178 $m^2$. As the 'existing state,' we used interpolated river bathymetry (using the triangular interpolation method on the measured cross-section data of BWDB), from the year 2020, as well as SRTM topography data (where necessary). The boundary discharge and water level can be seen in Section 4.

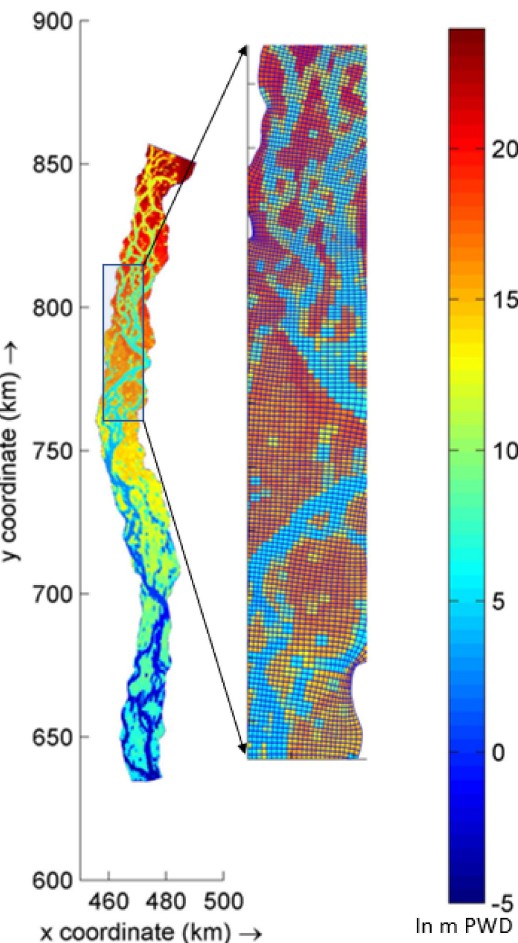

**Figure 3.** Model Grid and Bathymetry (right).

### 3.2.1. Hydromorphic Model Validation

The hydromorphic model was validated for the year 2011. Water level calibration was carried out at four locations: Chilmari, Kazipur, Sirajganj, and Mathura, while discharge and sediment calibration were carried out solely at the Bahadurabad station. (Locations are shown in Figure 1). Please see Supplementary Section S1.2.2 for the detailed validation of the hydromorphic model.

### 3.2.2. Assessment of Hazard

The climate-induced flood hazard, or any, should be evaluated based on statistical descriptors (i.e., flood severity), spatiotemporal descriptors (i.e., flood magnitude), and on socio-economical descriptors (i.e., the extent of flood damage, human casualties, and psychological impact) [1]. We consider the hazard as a nonlinear combination of flood depth, velocity, duration, sedimentation, and severity expressed as Equation (1).

$$H_z = \left( w1 * d_f + w2 * v_f + w3 * d_{uf} + w4 * s_{tf} \right) * Sev \qquad (1)$$

Here $w1$, $w2$, $w3$, and $w4$ are the weights for the hazard elements of depth, velocity, duration, and sedimentation. It is determined by using principal component analysis [56]. Table 3 shows the considered weights for different hazard elements.

**Table 3.** The values of the weights of the hazard elements.

| Weights | Value |
|---|---|
| Depth, $w1$ | 0.29 |
| Velocity, $w2$ | 0.27 |
| Duration, $w3$ | 0.32 |
| Sedimentation, $w4$ | 0.12 |

$d_f$ represents the *Flood Depth*, $v_f$ denotes *Flood Velocity* over the char, $d_{uf}$ stands for *Flood Duration*, and $s_{tf}$ is the *sedimentation* thickness over the char. The value of $d_f$, $v_f$, $d_{uf}$, and $s_{tf}$ are derived from the model results. *Sev* represents the flood *severity*, which identifies how unusual the flood or discharge [4] was [57]. Extreme flooding, recurrent untimely flooding, or prolonged flooding are often described as unusual floods by the local char community [58]. It is not an exact descriptive statistic. We followed the method used by Brakenridge [4] while assessing the flood severity of large river floods across the globe. Though he used the method only to quantify the peak characteristics, we extended the analysis along the entire hydrograph. Based on the flood recurrence interval or return period, three flood severity classes were established: large flood occurrences with a return period of 10–20 years are classified as Class 1, very large flood events with a return period of 20–100 years are classified as Class 2, and extreme flood events with a return time equal to or greater than 100 years are classified as Class 3.

Several flood hazard indexes, ranging from 0 to 4, are assigned based on local people's perceptions, literature review, and model outcomes, as described in Table 4. A hazard ranking of 0 indicates a very low hazard corresponding to the inundation depth range of 0–1 m, overland flood velocity of 0 to 0.58 m/s with an average flood duration, which is 13 days, and sedimentation thickness considered from 1 to 2.7 m. The sedimentation thickness below 1 m is deemed to be essential for the char land formation process [38,59]. Accordingly, rank 4 implies a very high risk of catastrophic harm to life and property, which corresponds to flooding depths larger than 4.5 m, high velocity (>0.8 m/s), and 100 days with sedimentation greater than 3 m.

**Table 4.** Description of the Hazard scale used in this study.

| Depth (m) | Duration (Days) | Velocity (m/s) | Sedimentation (m) | Hazard Ranking | Hazard Zone | Definition of the Hazard Zone |
|---|---|---|---|---|---|---|
| <1 | <13 | < 0.58 | 1 to 2.7 | 0 | Very low | Causalities and property damage is expected to be the lowest |
| 1 to 2 | 13 to 45 | 0.58 to 0.62 | 2.7 to 2.8 | 1 | Low | Causalities and property damage is expected to be very low |
| 2 to 3.5 | 45 to 70 | 0.62 to 0.67 | 2.8 to 2.9 | 2 | Medium | Causalities and property damage is expected to be relatively higher |
| 3.5 to 4.5 | 70 to 100 | 0.67 to 0.8 | 2.9 to 3 | 3 | High | Property damage is extensive, and the likelihood of causalities is high. |
| >4.5 | >100 | >0.8 | >3 | 4 | Very high | At all levels, severe damages are expected |

There is no inventory of flood damage to the Char region of Bangladesh. Therefore, community members' perceptions of flood damage were linked to these scores. The flooding of 1994 or 2006 is represented by a hazard ranking of 0. The flooding of 1999 or 2011 is represented by rank 1. The flooding of 2000 or 2008 is represented by rank 2. Flooding in 1997 or 2014 is represented by rank 3. The flooding of 1988, 1998, or 2020 is reflected by rank 4 (see recent flood details [60]).

## 4. Results

### 4.1. Future Scenarios

For future climate simulations, the SWAT model has been simulated with the daily precipitation and maximum/minimum temperature data of three different realizations—driest (r2i1p1), moderate (r1i1p1), and wettest (r7i1p1) of EC-EARTH3-HR. The annual hydrographs of each of the periods—the 2020s (2006–2035), 2050s (2036–2065), and 2080s (2066–2095)—have been prepared. Since climate data contains outliers, extreme flood scenarios are usually defined by the 90th percentile instead of the maximum of the climate dataset (Roy et al. [19]). In this study, the 90th percentile annual daily flow hydrograph of Brahmaputra is developed for each of the realization periods. For the 2080s of RCP 8.5, the highest discharge is found to be 102,450 $m^3/s$, 98,881 $m^3/s$, and 83,464 $m^3/s$ for the wettest, moderate, and dried ensembles, respectively. To compare the results with the regular flooding event, another flow hydrograph is considered with a return period of 2.33 years, as shown in Figure 4. These ten flow hydrographs are later used as the flow input for the Delft3D model. As mentioned earlier, in the case of downstream boundary, water level hydrographs are generated using the rating curve of discharge hydrographs, as displayed in Figure 5. For every condition except the average condition, the adjustment of sea-level rise was made using the base level adjustment mentioned in ref. [26] (shown in Table 5).

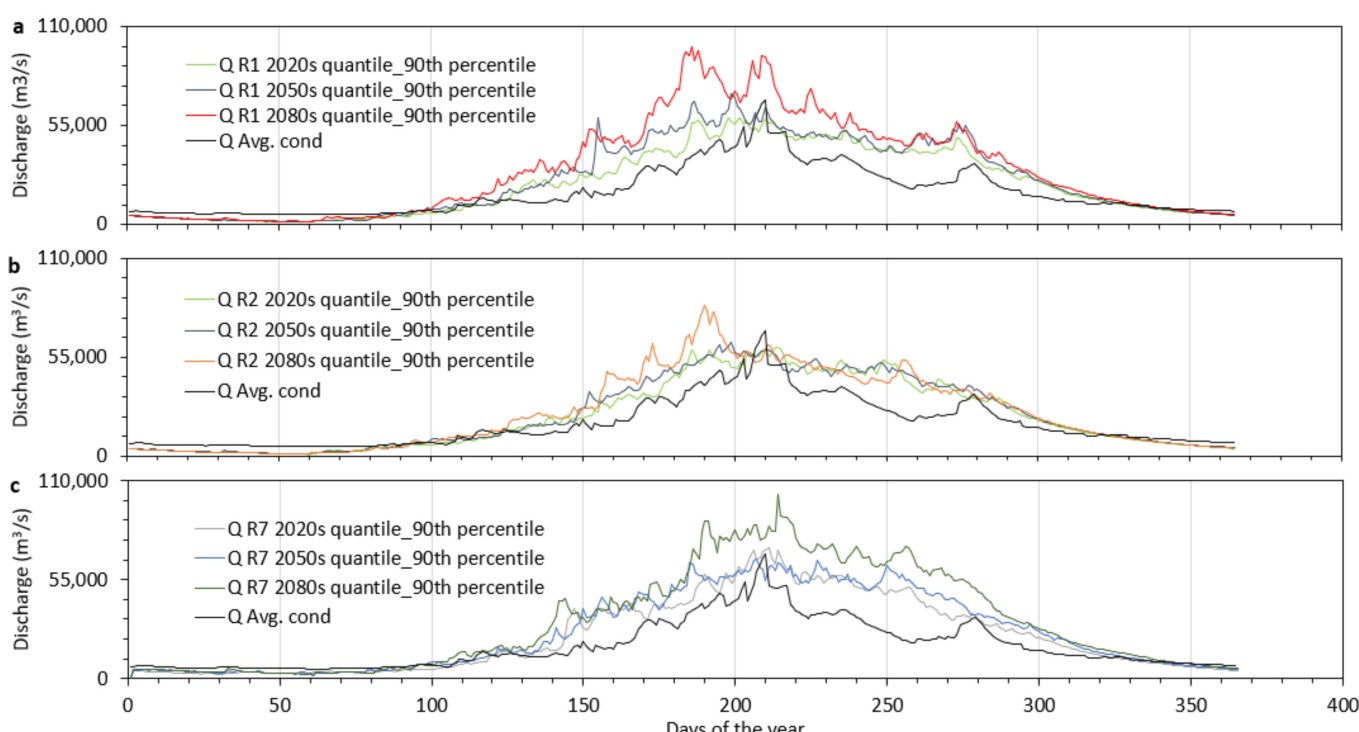

**Figure 4.** The 90th percentile annual Daily flow hydrographs for 2020s, 2050s, and 2080s for (**a**) Moderate (R1i1p1), (**b**) Driest (R2i1p1), and (**c**) Wettest (R7i1p1).

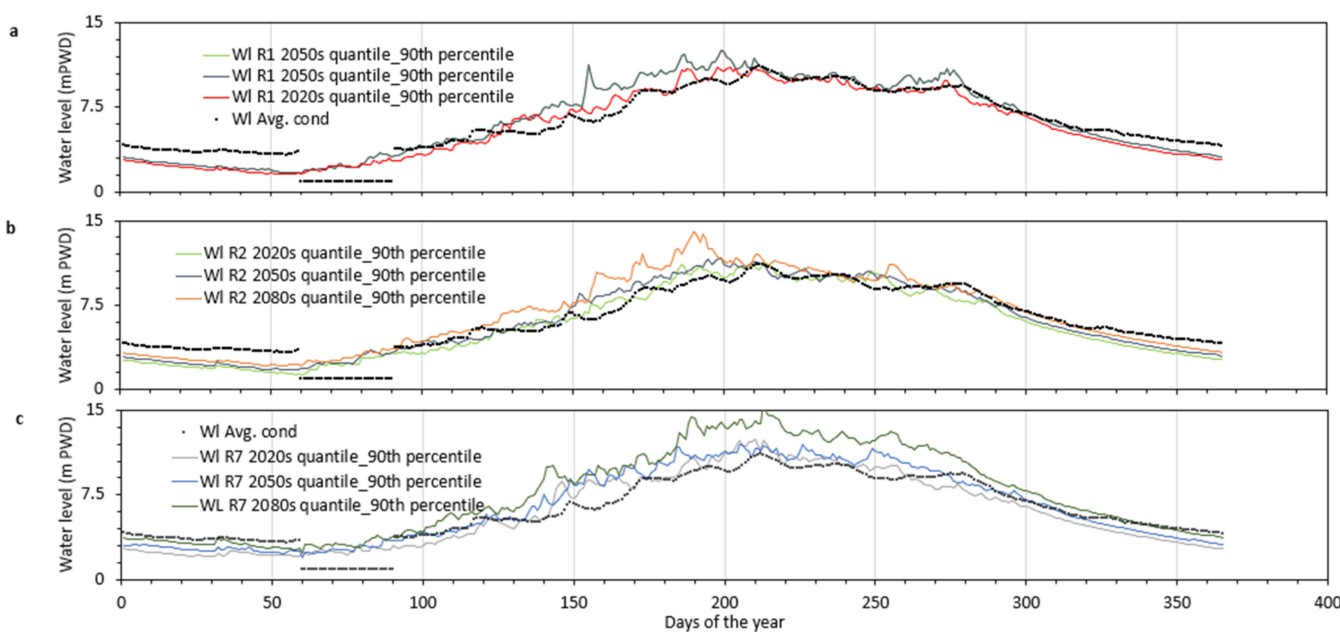

**Figure 5.** Water level hydrographs for 2020s, 2050s, and 2080s for (**a**) Moderate (R1i1p1), (**b**) Driest (R2i1p1), and (**c**) Wettest (R7i1p1).

**Table 5.** Adjustment of base-level at boundary conditions due to Global Mean Sea level rise.

| Condition | Scenarios | Global Mean Sea Level Rise (m) | Base Level Adjustment at Aricha (m) |
|---|---|---|---|
| Sea level rise RCP 8.5 | 2020s | 0.17 | 0.16 |
| | 2050s | 0.38 | 0.37 |
| | 2080s | 0.82 | 0.81 |

### 4.2. Flood Severity

Flood severity is a measure of the extremity of flood or discharge. Brakenridge [4] estimated the severity during the peak flooding time to measure the unusuality of the peak flood. Figure S17 shows an example of flood severity of an observed extreme flood event in the study area. Here, we used the same procedure for the whole hydrograph to calculate the severity of the future scenarios. Figure 6 shows the relationship between the return period and the severity of the considered future scenarios. In the case of R1 scenarios (moderate condition) in the near-future case (the 2020s), the return period of the hydrograph varies from 1.03 years to 15.56 years. Therefore, the corresponding severity is 1. In the mid-century case (the 2050s), the return period varies from 1.01 years to 95.33 years. Hence, the maximum severity is 2. In the end century case (the 2080s), a very time-varying return period is observed, ranging from 1.05 years to 95.33 years. In this case, the severity varies from 1 to 2. For the driest condition, R2, the return period varies from 1.01 years to 95.33 years, with an average value of 3.33 years. However, the variability of the return period is less than R1 cases. In this case, the maximum severity was also 2, as the return period does not exceed 100 years. In the wettest case, the R7 similar pattern is true for near-future (the 2020s) and mid-century (2050s) cases. The end-century case (2080) shows very irregular severity, ranging from 1 to 2, having an average of 1.18.

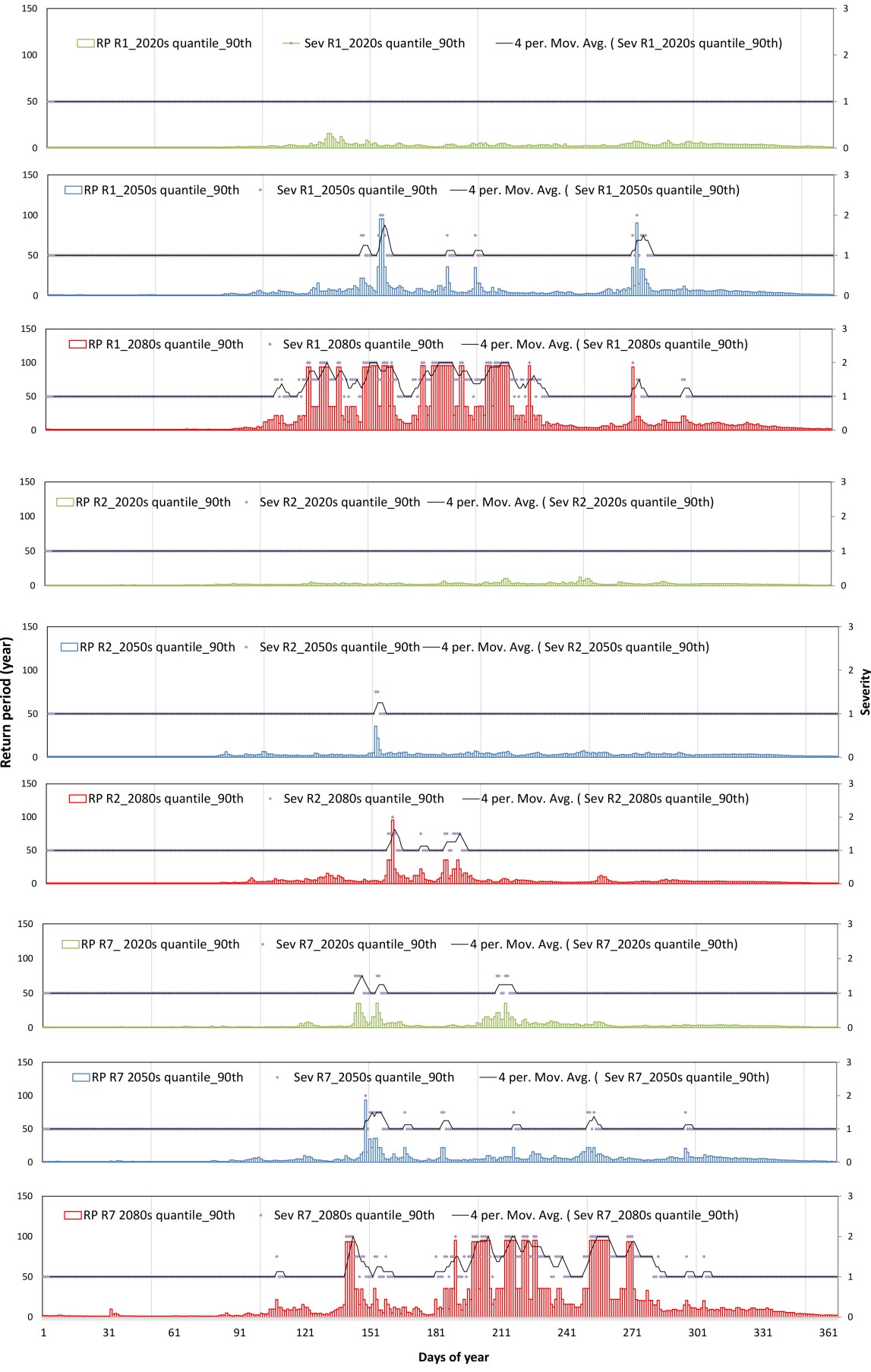

**Figure 6.** Return period and severity of the considered future scenarios.

### 4.3. Inundation

Figures 7 and 8 depict the inundation characteristics of the flood in average and future climatic scenarios. Figure 7 shows the flood inundation at peak flood time in different scenarios. Figure 8 presents the histogram plot (excluding the depth of channel) of inundation depth over the char of the considered cases. Here, the depth greater than 10 m represents the braided channels. From these two figures, it is evident that, in the average condition (used as a base condition in this study), the flooding depth varies from 1 m to 2 m over the chars. Near the char adjacent channels, the high (>5 m) inundation depth is observed (Figure 7a). In climate scenarios, i.e., R1 scenarios (moderate condition) (Figure 7b–d) in the near-future case (the 2020s), the overall depth is also found between 1 m and 2 m. In the mid-century case (the 2050s), the depth varies between 2 m to 3 m (Figure 8) with some high depth at the near channel as well. In the century case (the 2080s), the average depth is found to be 5.33 m, which is almost 87% higher than the average condition. In R2 (driest case) condition (Figure 7e–g), i.e., the near-future case, the average depth is found to be 3.08 m, which is 8% higher than the average condition. In mid-century cases, the maximum depth falls in 1m to 4m classes, with an average of 3.26 m (Figure 8). The R2 end-century case follows a similar tendency with an average value of 4.34 m, which is 52% higher than the average condition. In the R7 scenario (wettest case) (Figure 7h–j), in the near-future cases, the average inundation depth is found at 3.73 m. The condition worsens in the end century case (the 2080s) where the average depth is 5.02 m, which is nearly 83% higher than the average condition.

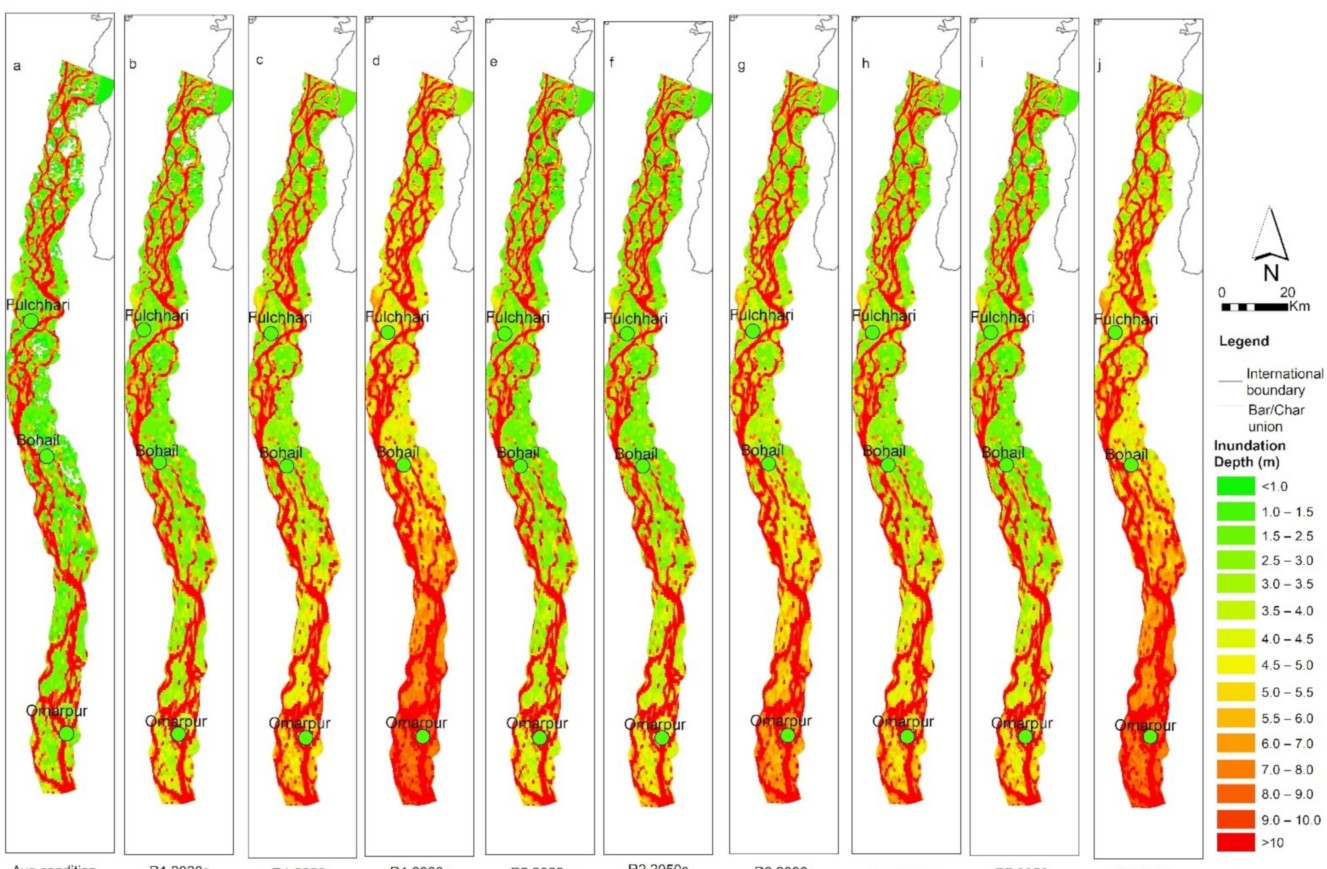

**Figure 7.** Flood inundation at peak flood time in different scenarios.

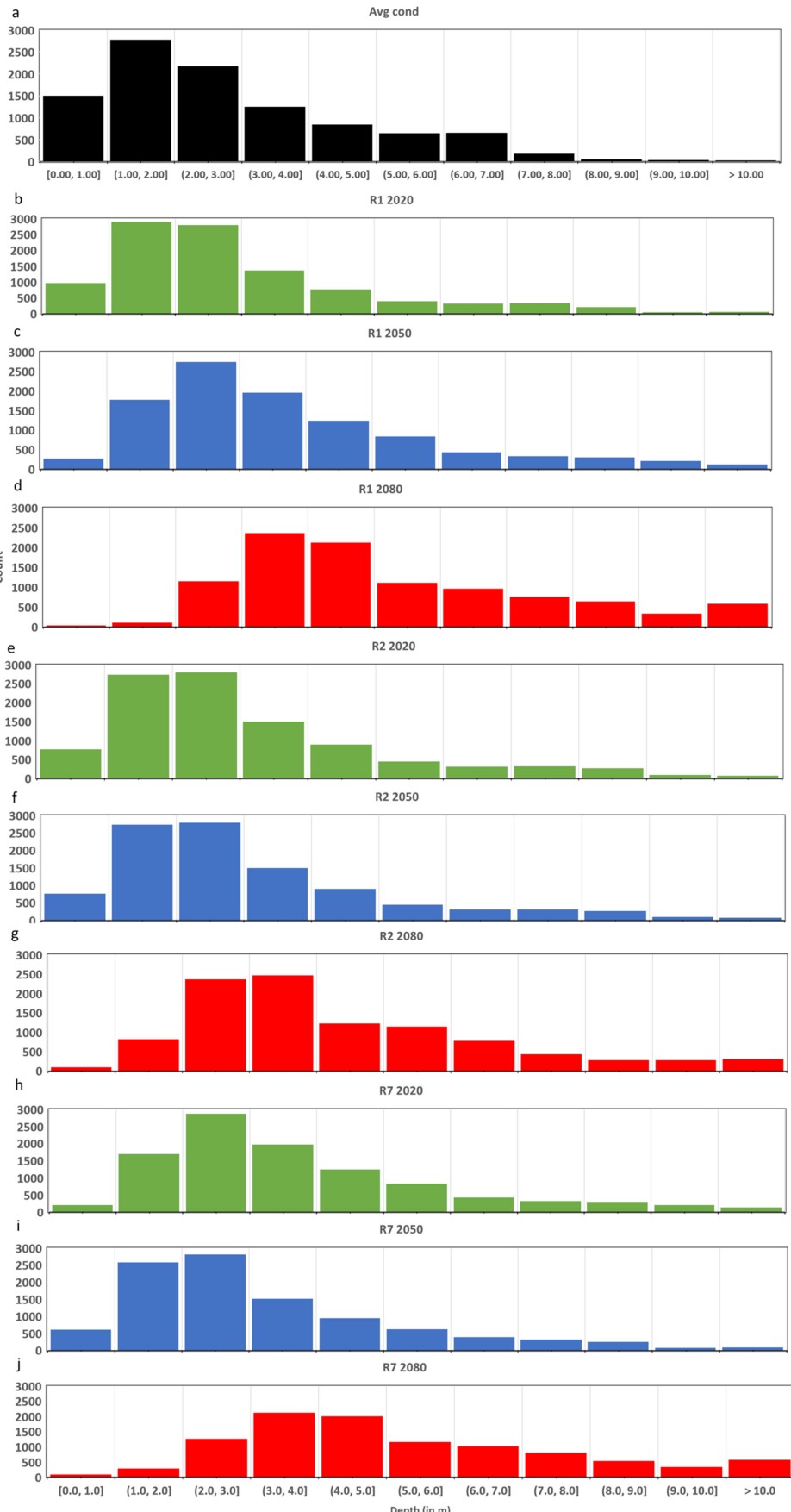

**Figure 8.** Histogram of inundation depth over the char of the considered cases.

It is also evident from Figures 7 and 8 that the downstream chars are mostly affected in climatic condition scenarios. For example, if we closely observe the inundation condition of three char union Fulchari (mid-channel bar), Bohail (attached bar), and Omarpur (mid-channel bar), Omarpur union is likely to be more inundated in all future scenarios (80% higher than the average flood condition) (Figure 9).

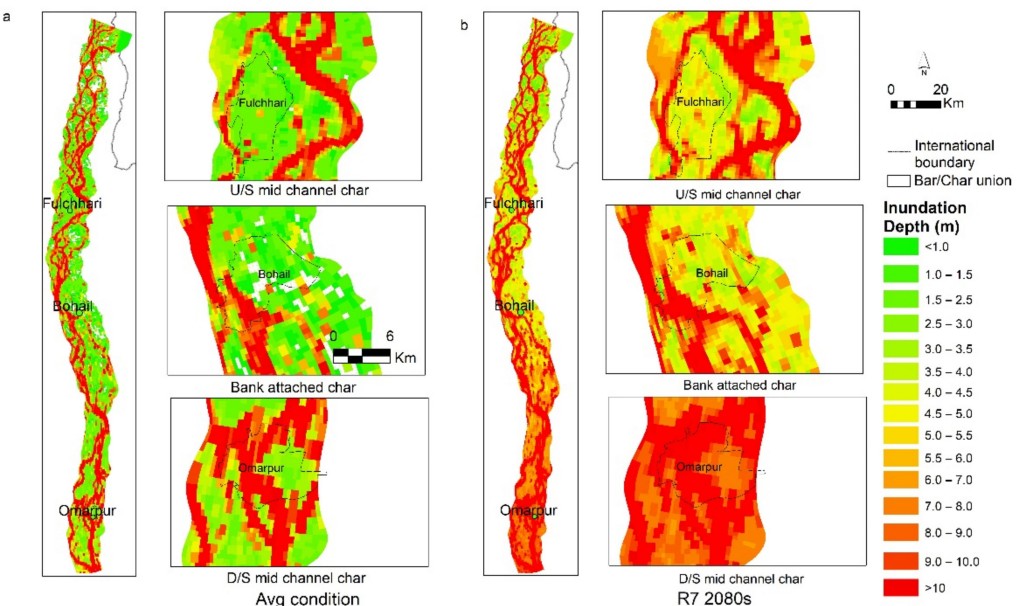

**Figure 9.** Variation of inundation in different chars.

### 4.4. Velocity

Flooding velocity is a space-varying variable, and along with the same char, a different velocity distribution is observed with the location. Therefore, we considered the spatially averaged depth-averaged velocity (depth-average-velocity over the chars during the peak discharge time) while comparing the different scenarios. In average flooding conditions, the depth average velocity is found 0.55 m/s over the char (Figure 10). In future climatic conditions, it varies between 0.55 m/s (R1 2020s) and 0.83 m/s (R7 2080s). End-century cases (the 2080s) may likely have higher velocity (53% than the average condition) in all conditions. In general, the velocity in upstream chars is high compared to downstream ones (Figure 11a). For example, during the average flooding condition, the velocity at Fulchari varies between 0.65 m/s and 0.90 m/s, while in Omarpur, it is from 0.03 m/s to 0.64 m/s. In future scenarios, it is multiplied by nearly 1.6 times during the R7 end-century conditions, but near the downstream end, the velocity is reduced due to the change of base condition. In the case of the Omarpur union, the velocity is reduced to 0.04 m/s in R7 2080s (Figure 11b).

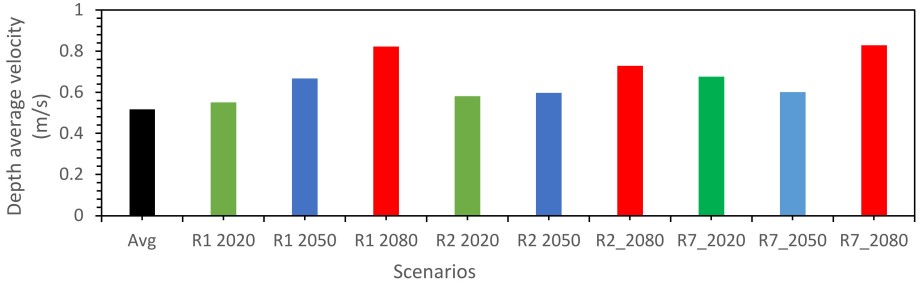

**Figure 10.** Depth Average Velocity during the peak flooding time.

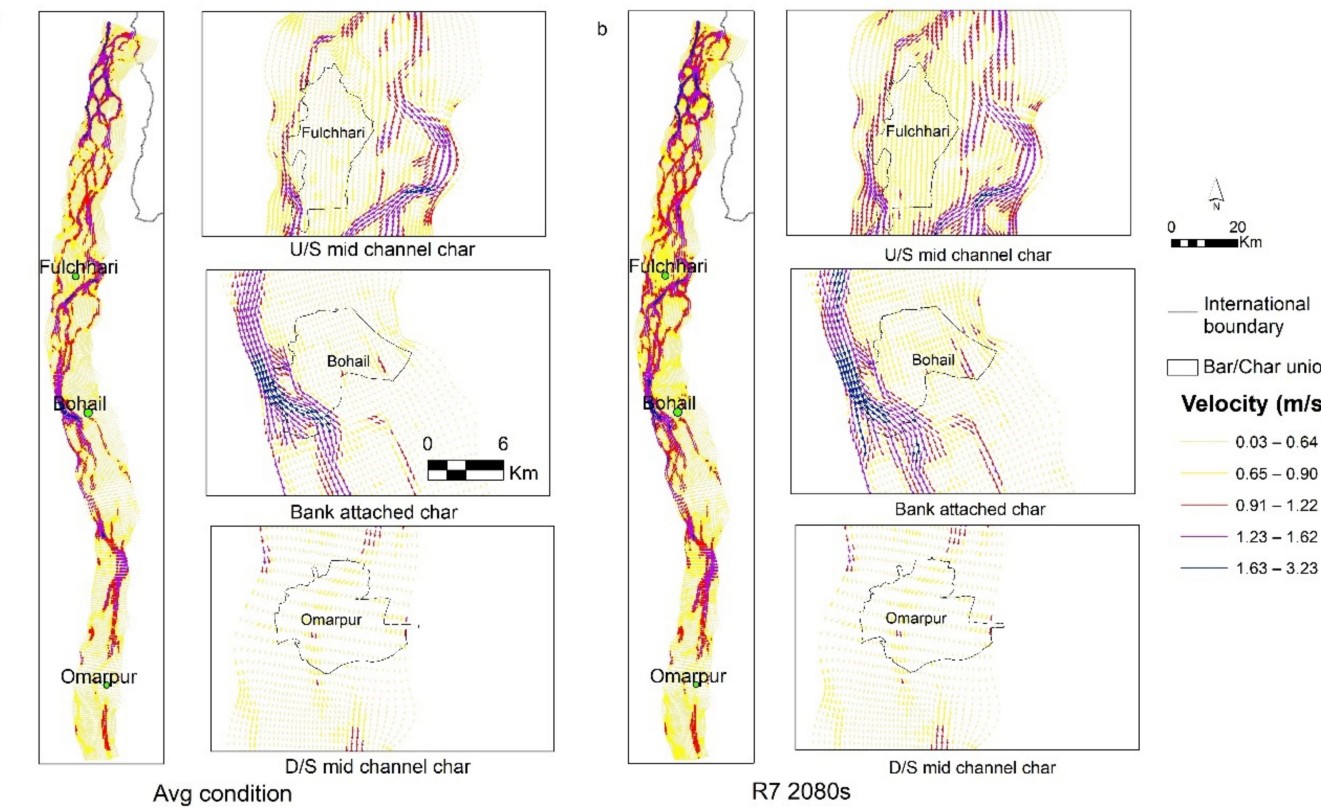

**Figure 11.** The variation of flood velocity along with the chars.

*4.5. Duration*

The chars of the Brahmaputra–Jamuna contain land of various persistence of age [61]. Normally, older chars have a higher elevation than the younger, and human settlements are more common on those elevated lands. We calculated the flood duration grid-cell-wise and considered the maximum duration of continuous water existence as flood duration throughout the simulation period (one year). Figure 12 shows the maximum duration of the continuous flooding in the study area, whereas Figure 13 shows the box plot flood duration for different scenarios. Flooding duration is less in older chars compared to the low-lying areas, as shown in Figures 12 and 13. Therefore, we consider the interquartile range (between first and third quartiles, value ranges 25% to 75%) in explaining the data, excluding the flood duration in very lowlands where it should not be treated as a hazard. A wide range of flood duration is observed due to the variation of land elevation. In average conditions, the maximum continuous flood duration varies from 1 to 13 days. A longer duration is observed in the downstream chars (i.e., Omarpur union). With the increase in discharge in several climatic conditions, the flooding duration increases, i.e., in R1 2020, and most of the duration (interquartile range) varies between 35 and 87 days, while in R1 2080, it fluctuates between 62 and 117 days. In the driest condition (R2 scenario), it ranges from 24 (R2 2020s) to 103 days (R2 2080s). In the wettest condition, 58 to 76 days are observed. Mondal [39] found the duration of average flood to be nearly 14 days (on house plinth level) in the chars of Jamuna. In historical extreme floods, such as floods in 1988 or 1998, the duration was recorded as nearly 21 to 63 days [39]. In recent years (2020), the recorded flood duration was also nearly 63 days [60]. However, the past literature did not quantify the elevation level of chars by which they defined 'flooding'.

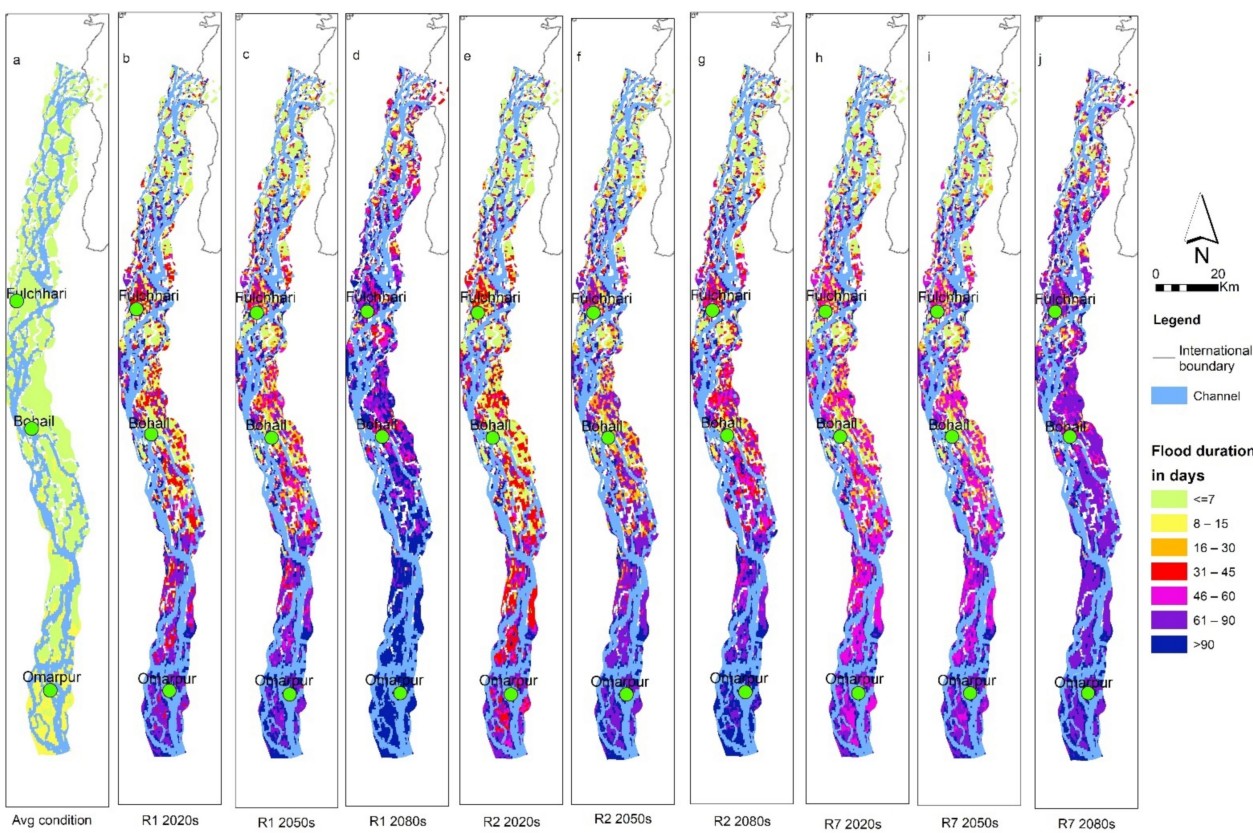

**Figure 12.** Continuous flood duration in the study area for considered climatic scenario.

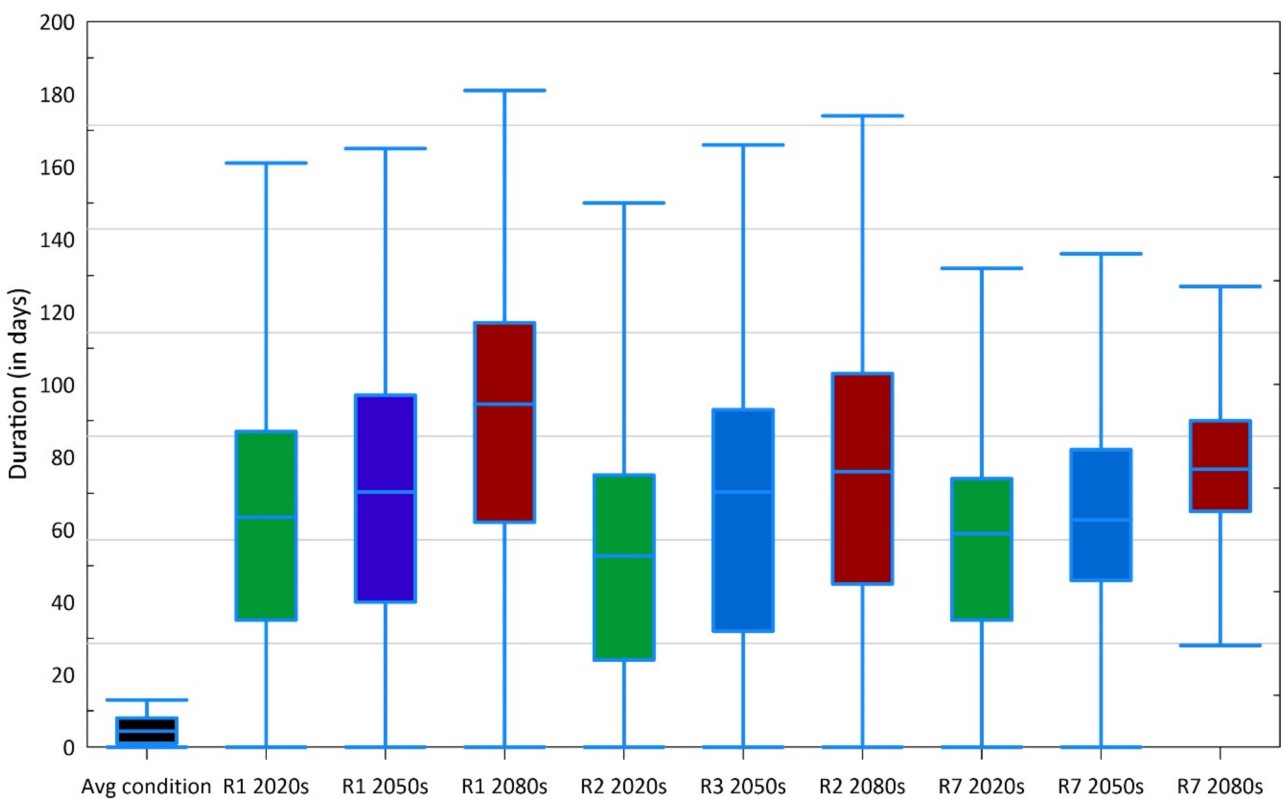

**Figure 13.** Box plot of flood duration for different scenarios.

## 4.6. Sedimentation

During the flooding time, the channels and the chars both experienced sedimentation and erosion. Erosion may lead to new channel formation and sometimes separates or erodes the bar completely [38,61]. Such erosion poses a different kind of hazard, which is beyond the scope of this study. We consider extreme sedimentation as an element of flood disaster as during a high flood, if the deposited sediment goes beyond the range of natural sedimentation range (greater than 1 m), the char dwellers need to suffer in the recovery stage of the flood. They need to flush out that sandy sediment before cultivation. Figure 14 shows the map of sedimentation and erosion after one flood. It is evident from this figure that the upstream chars are likely to have more extreme sedimentation than downstream chars. This tendency is true even for average flooding conditions (Figure 15). Fulcher and Bohail are likely to experience higher extreme sedimentation than Omarpur, even in one of the high-velocity scenarios (R7 2080). Figure 16 shows the box plot of extreme sedimentation in different scenarios. In average conditions, most of the extreme sedimentation varies between 1.8 m and 4.5 m (interquartile range), with a mean of 3.3 m and a spatial distribution of 87.9 km$^2$. In a moderate scenario (R1), the maximum extreme sedimentation thickness varies (considering the interquartile range) from 2.7 to 6.6 m (R1 2080) for 132.5 km$^2$. In the driest condition, the extreme sedimentation thickness is quite similar in all cases with a mean thickness of 4.5 m, with a spatial variation of 139.07 km$^2$ to 146.68 km$^2$. The highest range of char extreme sedimentation is observed in the wettest conditions (R7). Here, the maximum sedimentation varies from 2.9 m (R7 2020s) to 6.6 m (R7 2080), with a spatial variation of 151.66 km$^2$ to 223.87 km$^2$.

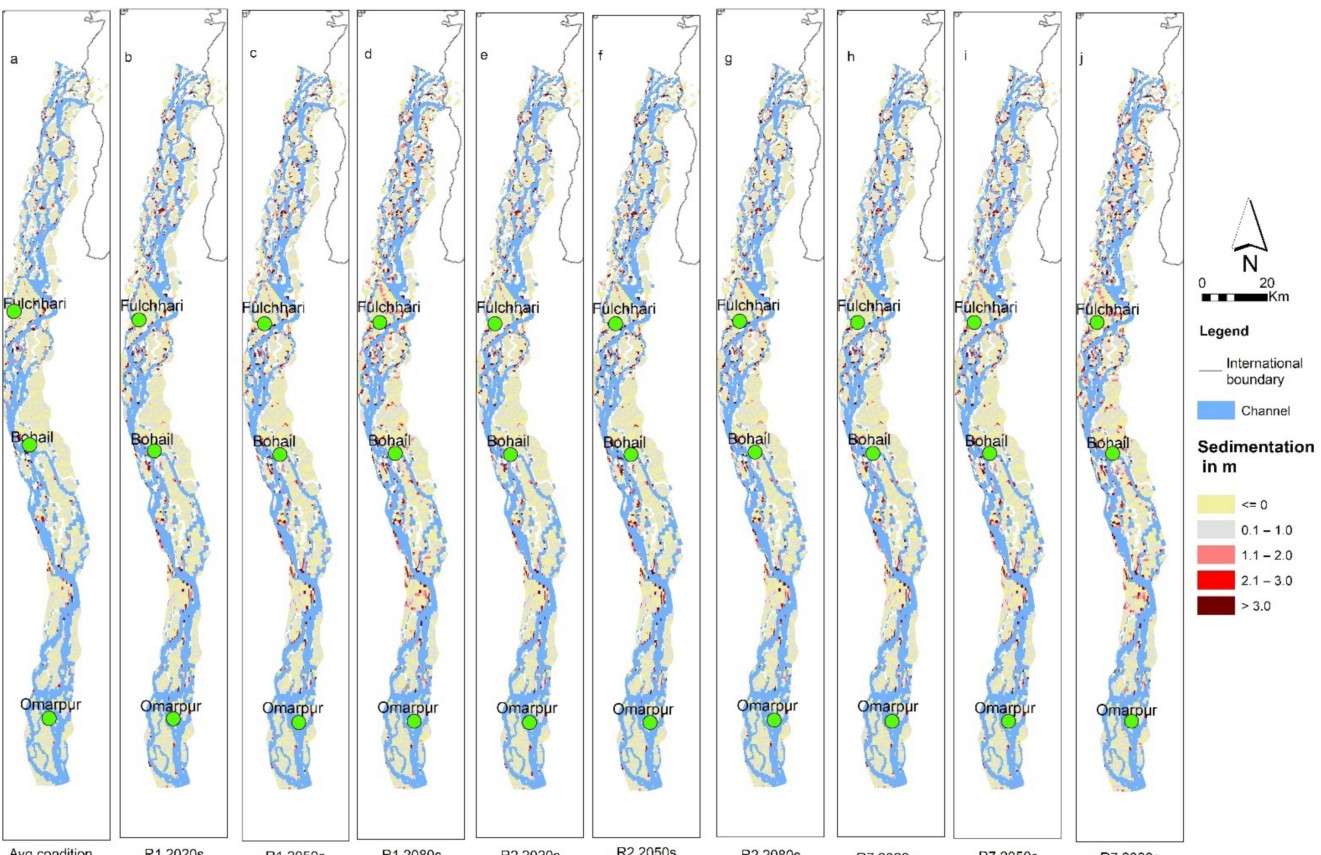

**Figure 14.** Mapping of the total sedimentation and erosion over the char of considered cases after one flood cycle.

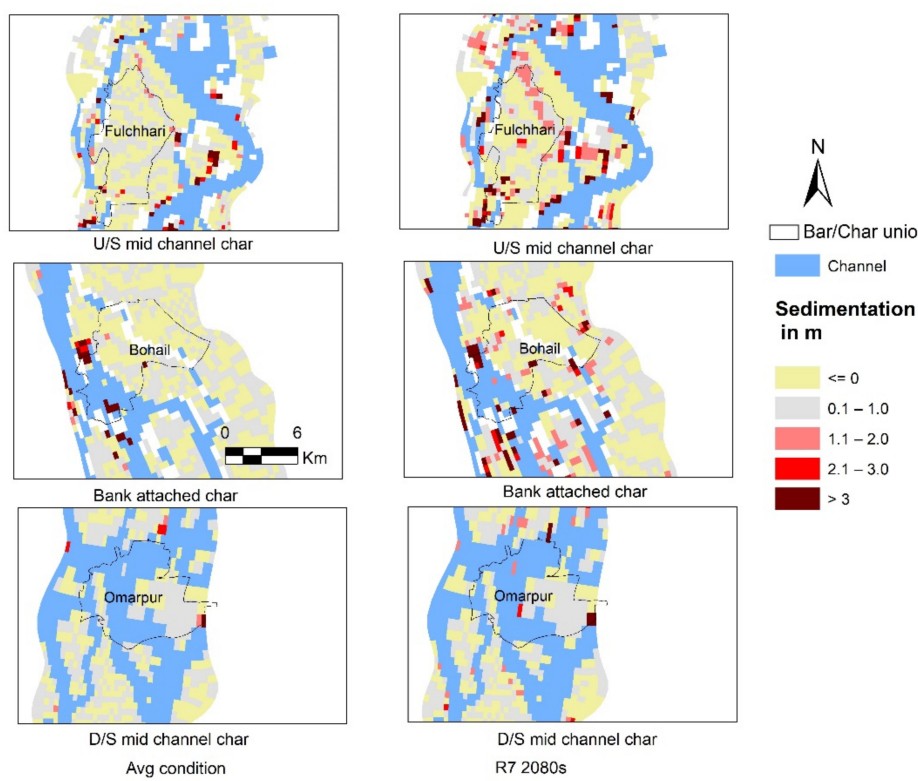

**Figure 15.** Sedimentation and erosion due to floods are shown at different locations of the char.

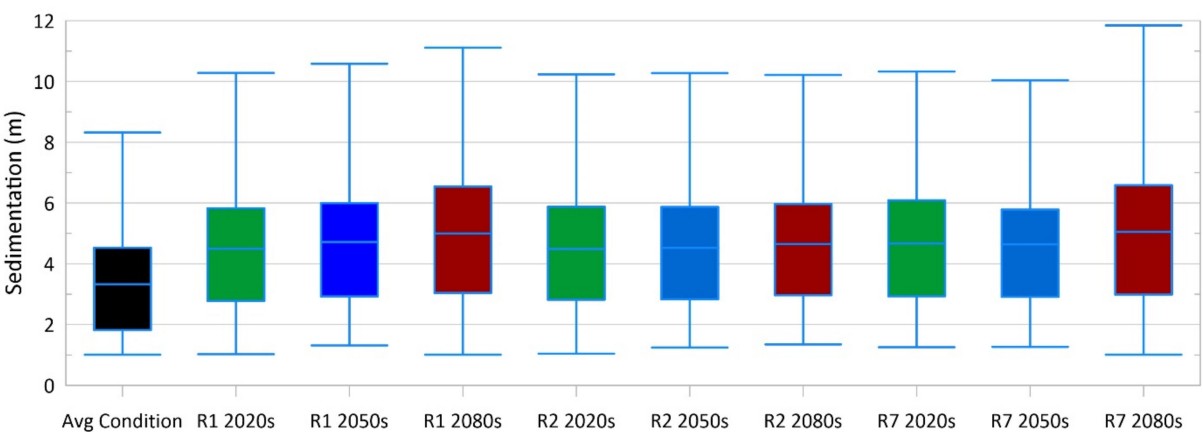

**Figure 16.** Box plot of sedimentation for different periods.

*4.7. Flood Hazard*

Using the time series analysis and simulation model results, the physical flood hazard maps for the different climatic scenarios are prepared and plotted in Figure 17. Flood depth, duration velocity, and extreme sedimentation are regarded as the core hazard parameters, which are multiplied according to the hydrograph severity. The flood threat in char grows, with time, from near-future to end-of-century in all climate scenarios, as shown in this figure. In average conditions, most of the chars show very low to low hazards. The maximum hazardous condition is found in R1 2080s, where almost half of the reach is experiencing very high hazards. Mid-channel char union shows very high flood hazard, while bank-attached char union shows low to no hazard condition, as shown in Figure 18. Among the mid-channel chars, the downstream char union (Omarpur) is more hazard-prone than upstream chars due to extreme base-level conditions.

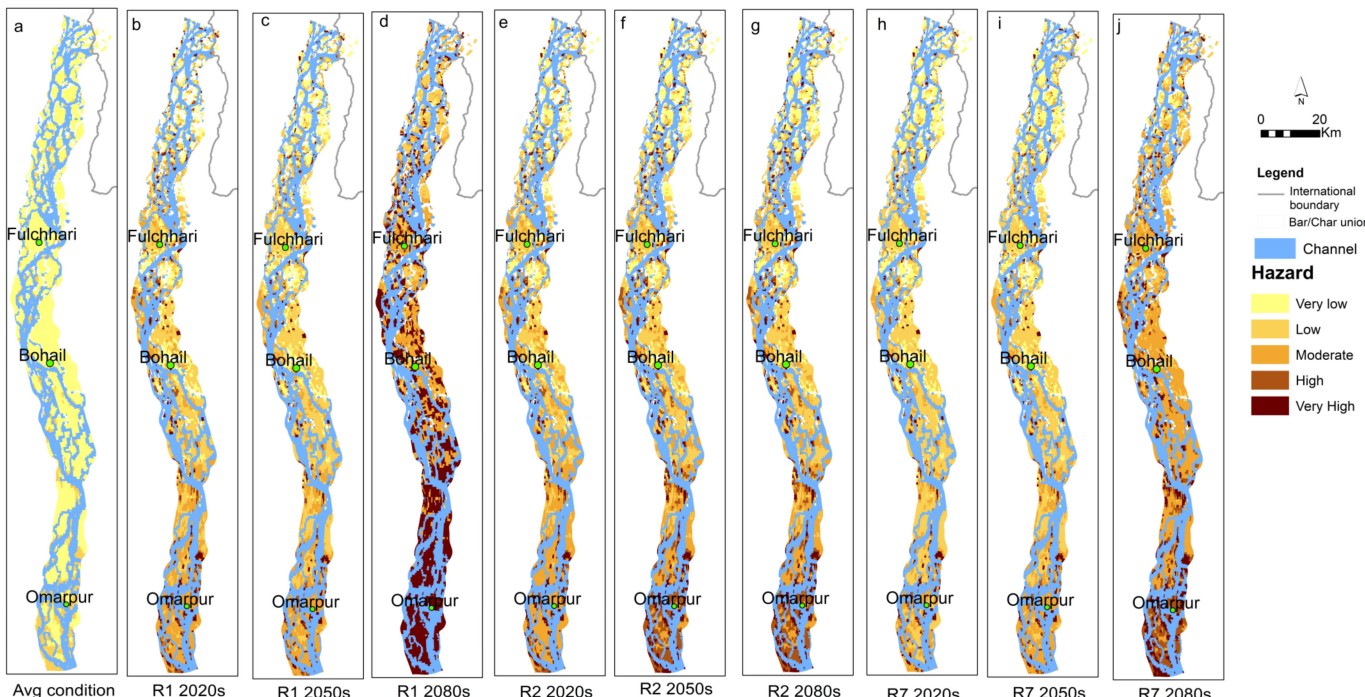

**Figure 17.** Hazard maps for different future periods.

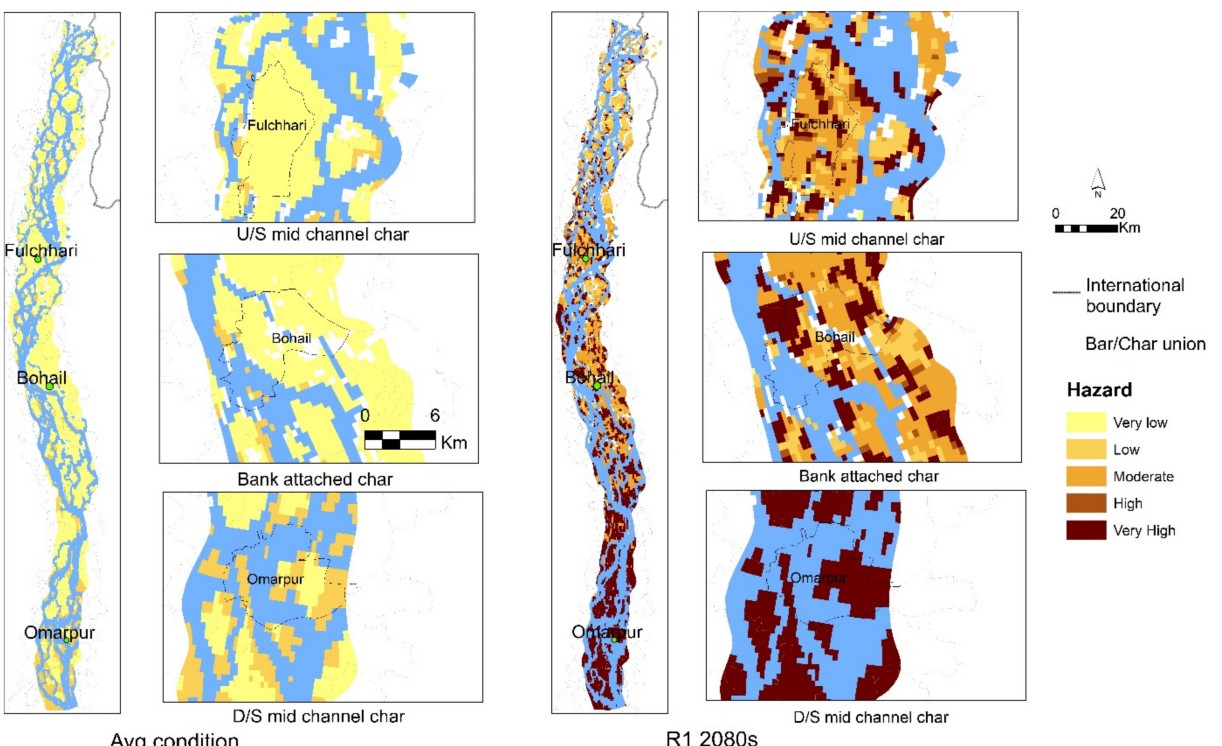

**Figure 18.** Hazard pattern for the upstream and downstream of the mid-channel and bank-attached-char at present and end of the century.

## 5. Discussion

Due to climate change, it is expected to have a higher frequency of large-scale river floods [62]. This study tries to capture the possible range of resultant hazards, considering the wettest, driest, and moderate climate change conditions for RCP 8.5 in the chars of

the Brahmaputra–Jamuna River. In these chars, the extreme sedimentation, as well as flood hydrograph unusuality, possess a substantial threat of flood inundation, duration, and velocity because of their agriculture-dependent life and livelihoods [39]. The climate-induced floods are compared with the observed average flood condition, and in all cases, the rivers' morphological responses are considered.

The char areas experience flooding every year. In Bangladesh the flooding in chars is not monitored exclusively, but we can consider the extreme flooding in the Brahmaputra-Jamuna basin as the extreme flood of the char areas as well. Figure 19 shows the Extreme value (EV-Gumble) distribution of the peak flooding (full hydrograph is shown in Figure 1) fifty-year period. Comparing these figures, it can be said that the floods of 1974, 1987, 1988, 1998, 2004, and 2007 can be considered extreme floods for this region. Previous studies showed that, during the flooding of 1998, 68% of the country was inundated with the loss of 9.75% Growth Domestic Product (GDP-normalized) [60,63]. During that time, the recorded flood discharge was observed to be 100,308 m$^3$/s, whereas the average flooding (2.33 per year return period flood) discharge was nearly 67,000 m$^3$/s (the year 2000 or 2002). Therefore, in the observed condition, 33% higher discharge can be experienced during extreme flooding compared to the normal one. Comparing the climate-induced extreme events (Figure 4) with the average conditions, the flow increased by 42%, 20%, and 47% (R1, R2, and R7, respectively), which can be comparable with the observed conditions. In climatic extreme conditions, the higher peak was observed in R7 2080s (102,450 m$^3$/s) and R1 2080s (98,881 m$^3$/s), and the nearest peak in observed data was 100,308 m$^3$/s, which occurred only once (1998). Among the 2020s, R7 conditions appear to have the highest peak discharge (72,852 m$^3$/s); 12 observed conditions had discharges greater than that. The R1 condition (72,747 m$^3$/s) generates the highest discharge among the 2050s, and 12 observed conditions were found to be greater than that.

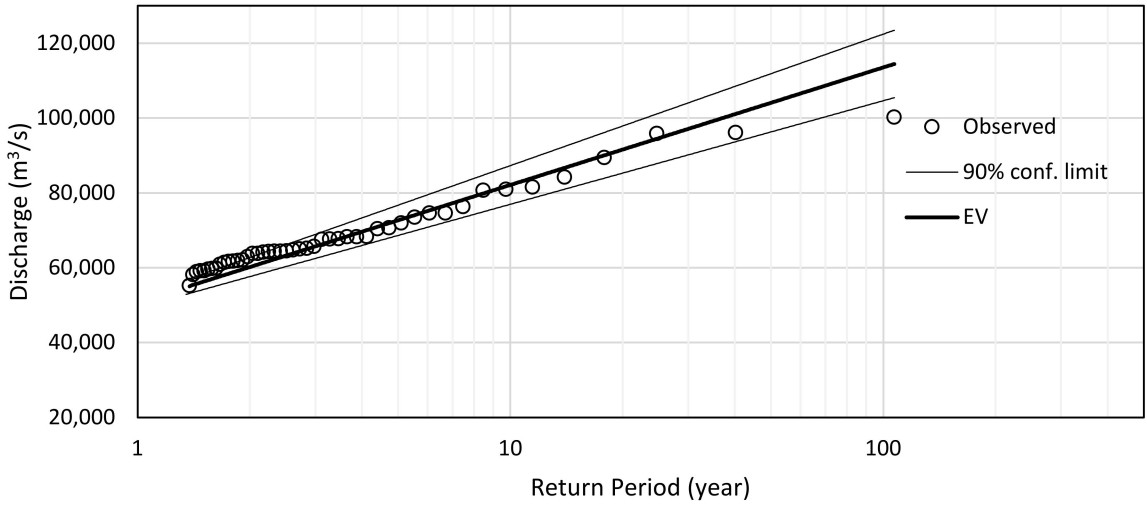

**Figure 19.** Extreme Value (EV-Gumble) distribution of peak floods of Brahmaputra–Jamuna.

During the flooding time, all chars are expected to be inundated to some degree; therefore, the inundation extent is not considered here. In all conditions, the depth, duration, velocity, and sedimentation increase with the progress of time from the near future to the end century (except for extreme sedimentation in R2 scenarios). These findings are consistent with the observations of a recent IPCC report [64], as higher precipitation is expected due to the high carbon emission scenario at the end-century time. Hence, the impact of climate change can be extreme by the end of this century for RCP 8.5. Shrestha et al. [21] also observed a similar increase in a hazard during the 2080s for the RCP 8.5 scenario in the case of the Yang River Basin, Thailand. Nevertheless, simultaneously, sea-level rise will increase, which will aggravate the downstream flooding condition, as shown in Figures 7, 12 and 17. Therefore, the downstream chars will be more affected. This type

of response is also observed in other downstream rivers (i.e., Arial Khan) of the country, as Roy et al. [13] mentioned. On a global scale, Abadie et al. [20] also showed higher city damages along the Bay of Bengal coastline in the 2070s.

The flood hazard maps of different conditions (Figure 17) also indicate that the end-century moderate climatic condition (R1 2080s) will be more hazardous, although the peak discharge (98,881 m$^3$/s) is lower than that of the wettest conditions (R7 2080s). The channel development process of the braided river may play a key role in that. In R1 2080s, the dry season flow was lower than R7 2080s (Figure 20a). Moreover, in R1 2080s, the first peak comes earlier than R7 2080s. Therefore, in R7 2080s, the braided channels get more time to adjust the cross-sectional area. As a result, the deeper channel is observed in R7 2080s. Such channel deepening affects the nearby area's flooding pattern, reducing the depth and duration of the flood. In contrast, the shallowing of the channel is also observed, which will eventually increase the hazard intensity. An example is illustrated in Figure 20b, wherein bed elevation near Bahadurabad station got a greater flow area for the same discharge (76,061 m$^3$/s) as of R1 2080s. Shrestha et al. [65] also investigated the morphological impact of climate-induced floods in the Chindwin River Basin, Myanmar. They concluded that sedimentation and river morphological changes are mainly due to high flood events, with no impact of low flow. In their case, however, the river type was meandering, for which the channel adjustment process takes a longer time than the braided river. Their simulation period of morphological assessment was also less compared to this study. Slater et al. [66] observed that changes in flood hazards caused by channel capacity were smaller but more frequent than those caused by streamflow while analyzing flood hazards in the United States, which is consistent with our findings.

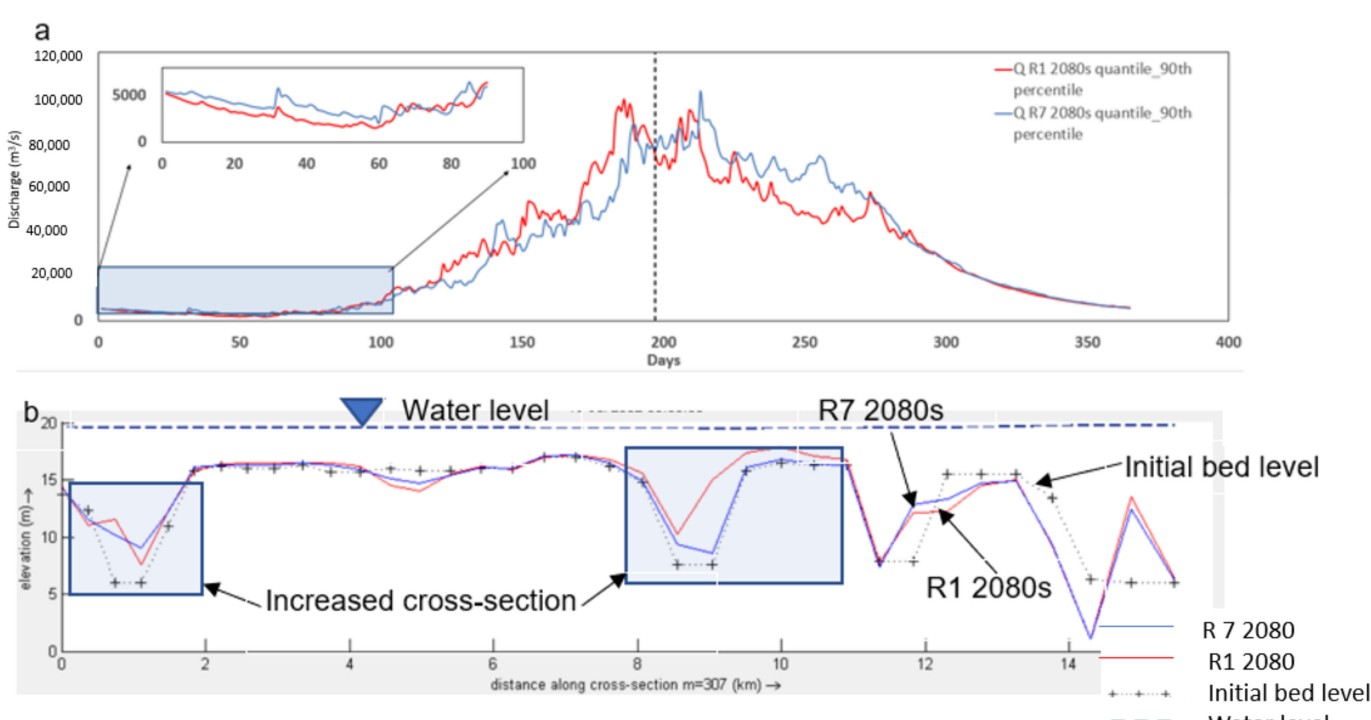

**Figure 20.** Discharge and Bed level change for R1 2080s and R7 2080s scenarios. (**a**) Discharge hydrograph (**b**) Bed level change near Bahadurabad station for R1 2080s and R7 2080s scenarios at the discharge of 76,061 m$^3$/s.

One limitation of the flood maps presented in this study is that the impact of sea-level rise at the downstream boundary of the hydromorphic model is estimated using a 1D modeling approach from the IPCC AR5 sea-level rise projections. However, the downstream boundary is nearly 300 km away from the river, and it is likely to be impacted

by the behavior of the connecting rivers that are ignored in this study. As the duration of the flood is likely to increase, the peaks of the other downstream rivers (i.e., Ganges) may synchronize with the Brahmaputra–Jamuna floods. This may worsen the flooding condition. As this study is focused on char areas, the exchange of flood water in the braided plain and the mainland was ignored.

Despite such limitations, this study attempts to generate flood hazard maps for char land, which may be useful as a soft measure of flood risk management. At present, the right bank of the river is protected by an earthen embankment (Brahmaputra-Right Embankment, BRE) to some extent. However, the protection from future extreme flooding by the construction of embankments in such a mighty river may not be fully successful. Ferdous et al. [67] show that such structures may increase flood risk. The breaching of BRE during extreme flooding is also quite common and needs huge maintenance costs [67]. As mentioned earlier, these nearly un-intervened chars are naturally resourceful. Therefore, future planning should focus on increasing their ecosystem services. Structural risk management measures are not feasible against hazards such as floods in chars; instead, some non-structural measures are suggested where the finding of this study can be instructive. In sensitive cases, where the river reaches are extremely aggressive to erosion, especially in the banks, a combination of structural and non-structural measures should also be adopted.

A relatively stable supply of sediment is required to maintain a consistent bed level. However, the supply of fluvial sediment in our study area is affected by changes in upstream catchments induced by climate and land use change, particularly reservoir/dam construction. Dunn et al. [68] showed that the Ganges–Brahmaputra–Meghna (GBM) delta's sediment flux decreased dramatically over time, from 669 Mt/a in a 'pristine' world to 566 Mt/a in the 'recent' past and to 79–92 Mt/a by the end of the 21st century, with a total average rate of decline of 88%. This reduced sediment supply may influence the river morphology. Because the river is alluvial, the sediment-deprived flow may attempt to achieve equilibrium by removing sediment from the bed and banks. In that case, the chars' existence could be jeopardized.

## 6. Conclusions

Though extreme flood events occur rarely, they are considered one of the most devastating natural hazards. Flooding is a catastrophe, but the catastrophe can be used wisely to improve risk management strategies. Climate change is supposed to induce a global threat by increasing the frequency and intensity of such extreme events. Therefore, we need to expand our understanding of extreme flooding processes and hazard assessment approaches for flood-prone places. We consider the severity of extreme floods, as well as the hydro-morphic responses of the river while assessing the flood hazard, using basin-scale hydrologic and reach-scale hydro-morphic simulations under RCP 8.5 scenario in three timelines—near, mid, and end centuries. We found that the effects of such extreme events are relatively small in near-future but likely to have a severe impact by the end century, which is true for all conditions moderate, driest, and wettest. The flood severity may manifold two times compared to regular flooding. The inundation depth may increase two to three times more than the regular flooding in the mid-century, but the situation may worsen in end-century times when extremely high flooding depth (3 to 4 times) is likely to occur. The depth average velocity may increase 1.5 times more than the average flooding condition. The upstream chars are severely impacted compared to the downstream chars due to velocity changes. We also found that the chars may experience long-duration floods, which would continue until November. Though the average flooding duration along these chars is nearly 45 days, it may go up to 120 days in extreme climatic conditions. During regular flooding events, the chars experience sedimentations less than 1 m, but they may increase three-fold due to climate-induced floods. The hazard maps uncovered that the highest discharge condition is not always possessing a severe hazard, as it depends on the hydrograph pattern. The river always tries to adjust its cross-sectional area with the upcoming flow; therefore, we found the most severe hazard in moderate

conditions (R1 2080 scenario), not in the highest peak flooding condition. The chars may be affected disproportionately based on their location. The downstream chars (mid-channel) may have greater impacts than the upstream chars in terms of flooding depth, duration, and sedimentation.

Although we cannot prevent extreme flooding, we can limit the consequence by using appropriate flood hazard assessments. Any flood risk mitigation measures in the chars of the Brahmaputra–Jamuna River should focus on optimizing their ecosystem services through the understanding of regular—as well as extreme—flood hazards and their consequences.

**Supplementary Materials:** The following supporting information can be downloaded at: https://www.mdpi.com/article/10.3390/geohazards3040024/s1, References [69–81] are cited in the Supplementary Materials.

**Author Contributions:** Conceptualization, S. and B.R.; methodology, S. and B.R.; software, S. and B.R.; validation, S., A.K.M.S.I. and K.M.; formal analysis, S., B.R., M.M.H. and M.A.R.; investigation, S. and B.R.; writing—original draft preparation, S. and B.R.; writing—review and editing, A.K.M.S.I.; supervision, A.K.M.S.I. All authors have read and agreed to the published version of the manuscript.

**Funding:** This research received no external funding.

**Conflicts of Interest:** The authors declare no conflict of interest.

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
