# Peer review of "Assessment of Flood Hazard in Climatic Extreme Considering Fluvio-Morphic Responses of the Contributing River: Indications from the Brahmaputra-Jamuna’s Braided-Plain"

_2624-795X, doi:10.3390/geohazards3040024_

Round 1

Reviewer 1 Report

Review of Assessment of Flood Hazard in Climatic Extreme Considering Fluvio-Morphic Responses of the River by Shampa et al.

This paper presents the results of a fluvio-morphic Deflt 3d model of a 225 km reach of the Brahmaputra-Jamuna river. The model addresses an interesting and important topic but in some sections lacks detailed explanations and scientific rigour. If the suggestions presented here are incorporated I think this will be an interesting paper worthy of publication in GeoHazards. 

General points:

1.     In general, more detail should be included in table and figure captions. Avoid use of abbreviations in figure legends, unless explained in the caption. 

2.     The paper title suggests that this study is focused primarily on fluvio-morphic induced flood hazard. From what I understood, the sedimentation component of the paper is a minor component of the paper and not presented or analysed with any significant detail. 

3.     It is not clear to me how the future forecasted scenarios can be compared to the 2.33 return period present day flood. If the future scenarios are obtained for the 90% percentile future flood events for each scenario, should they not be compared to a more extreme present day flood? For example, are the percentages given in Table 2, the percentage increase in rainfall when compared to a present day 2.33 year rainfall event? If so, this should be included in the text explicitly. More detail should be included into how each of the future flood scenarios compare to present day return intervals. There is a lack of discussion on present day return period floods, and past extreme flood events. Including a return interval plot (Gumbel or GEV, for example), and an historical times series hydrograph would help in this regard. 

4.     On a similar note, there is little description of past hazardous events in the region. What was the most severe flood which occurred in the past in terms of discharge and damage to people, property, agriculture etc? 

5.     In regards to the numerical modelling, it is not clear what the boundary conditions (BCs) are used along the sides of the domain (in the streamwise direction). BCs are only defined upstream and downstream. It would help to include a shaded DEM of the study region and immediate surroundings. In my own research I have not studied this region in detail but the model output presented in Fig 10 (flood inundation maps) suggest that the region is either boardered by high terraces (> 5 m high) or else there is an issue with the side wall BCs. Please clarify. 

6.     The difference between severity and return period is unclear. Is it the same thing?

7.     The DEM used is quite coarse resolution to investigate sediment dynamics. Comment on this in the paper. The Delft 3d mesh is even coarser. Did you do any grid convergence testing?  

8.     The near future value used is called “2020s”, which is really the present. I give some examples below how you could use this scenarios to validate the model against observed data. 

9.     What is the population of the study area? Are all of the chars used for agriculture or settlements? 

10.  In the figures of flow velocity, water depth and sedimentation, it is not clear what time stamp that these refer to. For the former two, is it the peak values or a particular time? My understanding is that the latter is after the model has run a complete flood cycle? 

Detailed suggestions

1.     Line 36: delete

2.     Lines 40-44: This is slightly confusion. How is a flood less hazardous because people are used to experiencing it? Do you mean that there is a chance of higher risk for the same magnitude/intensity flood in a region that is not flood prone? If so, please give some references to back up this statement. 

3.     Line 45-46: should this be economic impact? Reference this statement.

4.     Lines 54-80: This paragraph can be condensed to avoid repetition. Several of the studies consider the same flood descriptors, e.g. depth/velocity, depth/duration. Group the references together based on the similarities.

5.     Line 57: “element” – should this be elemental or primary?  

6.     Line 63: “compoents”

7.     Line 65: (2005) give reference not year

8.     Line 67: “flood” (not Flood)

9.     Line 67: delete “while calculating the flooding hazard”

10.  Line 73: “distributary”? 

11.  Lines 77-80: These statements are a bit vague. Are you saying that in this particular study location river morphology is not a concern, and suggesting that it is only regions that experience debris flows that experience modification of the hazards? 

12.  Line 84: delete “by both ends (“

13.  Line 86: what do you mean by “change the outfall of the river”?

14.  Line 92: what is “climate modification” in this context? 

15.  Line 92: I find the suggestion that “annual minimum flow may also increase” surprising. Do climate projects not expects wetter wet seasons and drier dry seasons? 

16.  Line 93: add “by” between the words “increase” and “up to 24%”

17.  Line 99: replace the first “category” with “that”

18.  Line 100: add “by” or “for” before 1.1%

19.  Line 102-103: “downscaling” what sort of downscaing are you referring to? E.g. temporal/spatial? 

20.  Line 106-107: “quantifying the severity” is a bit vague. Similar to point 6 in general comments. 

21.  Line 112: the definition of “chars” should appear earlier, where the term is first mentioned. 

22.  Line 120: should 3200 km be 3200 km2. If this value is the river length, please replace with the drainage area. Similarly with Line 123-124, a catchment unit should always be given in km2, otherwise use the term river length.

23.  Line 124: I think that you mean “Along the whole path” as opposed to “Throughout”. Also on this line, “almost braided” means that the river is not quite braided. Do you mean to say that the river is braided along almost all of it’s course? 

24.  Line 126: add “reach” between “The” and “reach length”.

25.  Line 127: km, not Km

26.  Line 127: per 80 m2, not 80 m.

27.  Line 128-129: Where are the 300 chars shown? It is not clear from Figure 1(a), as suggested. 

28.  Line 130: repetition, delete. 

29.  Line 131: “in recent years” statement is misleading. The floods in the 90s are similar to the 10s from Fig 1(b). I think you mean that in the last 20 years there have been more extreme events compared to the years pre 1990, or similar?

30.  Line 133: What are “other external drivers”?

31.  Line 136: explain what the “braided index” and “char land persistency” are if you are going to refer to them.

32.  Line 138: The modelling is not “coupled”. Replace “couples” with “links”

33.  Line 145: replace “assessing” with “estimating”

34.  Line 151-152: How do you know that the data from the RCP 8.5 scenario are comparable to the 2.33yr return period event? Do you this event as the basis of the future projections? Please clarify in the methodology section. 

35.  Line 154: You mention a rating curve but no rating curve figure is given. Please provide. 

36.  Line 155-156: Clarify that you use discharge at the inlet and water depth at the outlet (at least that is what I understand from later sections). Otherwise you suggest an over constrained model. 

37.  Line 158: See note on severity above. 

38.  Figure 2: State where discharge and water level are obtained. 

39.  Table 1: It is not clear whether the future presented are projected “average” conditions. See point 34 above. In RCP 8.5 scenarios, do 

40.  Line 171: How do you know that this approach addresses “all these limitations”? Please give more detail and references. 

41.  Table 2: Contains too many abbreviations which have not been defined or explained. What units are the numerical values? Percentage changed compared to the 2.33 yr flood? 

42.  Section 3.1.2: Have you tried to validate the use of these climate models? Because the near-future case refers to the 2020s, perhaps you could compare the climate model projections with rainfall data from the last 2-3 years to validate the use of these models?

43.  Line 209: BWDB – I could not see a definition of this in the text. Please avoid using abbreviations without defining them. 

44.  Figure 3: The first point on the x-axis in (a) and (b) looks like a typo. Please fix. 

45.  Section 3.2: Comment on the model parameters used in references 53, 54, as well as the accuracy of their results and how they validated the model. 

46.  Line 220: I think you mean “briefly”, not “shortly”

47.  Line 220: As far as I can tell, this is reference 54, a doctoral thesis. Has this model been presented in a peer-reviewed journal paper previously? If not you will need to include more detail on the model parameters and calibration in this paper. 

48.  Line 234: “the sediment transport”

49.  Page 9: Make sure that any parameters are given in parameters when described in the text, e.g. Line 237 “c”

50.  Line 247: rho f not defined

51.  Line 257: Explain the “morphological factor”. Did you calibrate/test this parameter?

52.  Line 277: Grid cell size is given as 201 x 178. This suggests a uniform grid, is that correct? From Figure 4 (b) the grid cell size looks smaller in some regions. If the grid size is not uniform, explain how it was generated in Delft3d. Please explain in more detail. 

53.  Figure 4 (a) is irrelevant because we cannot see what the grid looks like from this image. I suggest you delete it. 

54.  Line 277-278: What is the source of the “interpolated river bathymetry”? What is it interpolated from? What kind of interpolation did you use? 

55.  Line 278: What is the 20-year flow? 

56.  Line 290: Where was the sediment data measured? Show the location on figure 1. 

57.  Line 304: Add reference number after the reference name. 

58.  Line 314: See comment 37.

59.  Equation (3.9): define term Sev after the equation. 

60.  Line 318-319: What is Principal component analysis of a system?

61.  Line 326: complete sentence with reference.

62.  Line 326-331: delete - repetition. 

63.  Line 332: “In” Table 4, 

64.  Line 339-340: more detail is required to explain what an “unusual flood” means. 

65.  Table 4: Can you include examples of these types of events which have occurred in the region in the past and discuss the damage that the events for the range of hazard class caused. As it stands it seems like an arbitrary description. 

66.  Table 4: Have large floods with depth greater than 3.5 m lasted for longer than 100 days in this region in the past? 

67.  Table 4: There is not enough description of past floods in the region to justify the use of the values given in Table 3. As a example, what if a flood lasted just 10 days but have a flow depth of 5 m, or a flow velocity greater than 0.8 m/s, what category would that be in? 

68.  Lines 344-349: If you refer to a rating curve, you need to include a figure in the paper. 

69.  Section 4.1: How the discharge values given in this section compare to different return interval floods developed using historical data and a stationary approach? 

70.  Line 367: See point 68

71.  Line 367: In Figure 10, it looks like you are getting some backwater flow caused by the downstream boundary. Did you test the effect that the downstream boundary has for the model? Was the water depth obtained using the rating curve combined with the upstream boundary discharge values? If so, this could overestimate the downstream depth because the reach is over 200 km long, leading to an inaccurate downstream boundary that increases downstream flow depths. Please clarify.

72.  Section 4.2: The descriptions in this section suggest that “severity” is the same as return interval. If that is correct, please state very clearly where the severity is first introduced. 

73.  Figure 9: How would this figure look for some of the extreme historic flood events?

74.  Figure 10: It is very different to read this figure. Please use the whole page for it and change the color ramp ramp so that it is easy to see the channels even when the depth is greater than 5m. use larger font.

75.  Figure 10: Use larger font. Explain what you mean by “class” on the x-axis.

76.  Lines 405-410: Add new figure of the zoomed in regions to help with this explanation.

77.  Line 420: What do you mean by “average velocity”? It is spatial-time-depth averaged velocity? If so, over what time period is the velocity averaged? Is it averaged over the whole area, or do you find the average over each char, and that average those values?

Note: after this section line numbers are missing. I will use section numbers from now on. 

78.  Section 4.5: explain “interquartile” and “Q1”, “Q3”

79.  Section 4.5: When you describe “flood” in this section, are you referring to peak flood, or a point where most (maybe all?) of the chars are inundated? How do the durations given here compare with historical events?

80.  Figure 14: Use a wider range for the colour ramp. Increase font size. 

81.  Figure 15: What are the bars? Uncertainty bands? Explain in text and caption. It looks like there is not a lot of variation in the flood duration across scenarios. If that is also the same when you look at historical flood records, mention in the text. 

82.  Section 4.6: Given the title of this paper, it is expected that this is the most important section. I am confused by the results. In line 449 you state that sedimentation can exacerbate a flood disaster when it exceeds 1m. In your results, there is little if no examples of sedimentation greater than 1m. This is repeated in Figures 16 and 17 where the maximum value shown in the colorbar is 0.3m, suggesting that the impact of sedimentation on flood hazards is negligeable for all scenarios presented in this paper. Please clarify. 

83.   Figure 19/20: Are these results calculated using the selection process of Table 4?

There are no line numbers in the discussion. 

84.  The 3rd sentence of the discussion is confusing as it suggests that sedimentation contributes equally to the flood hazard as much as the other components, which is contradictory to your findings, see point 82. 

85.  Sentence 5 explains how you obtain Fig 19/20. This should be included in section 4.7

86.  “In all conditions, the depth, duration, velocity, and sedimentation increase with the progress of time from the near future to the end century.” This is not evident for R2 scenarios, where there is little difference between near and mid future. 

87.  Line 493-499: With what level of certainty can you make this claim? 

88.  Figure 21(b) lines do not have color legend. 

89.  Line 499-500: Reference (Louise) Slater et al 2015. 

Author Response

Thank you for your positive feedback. All your comments were extremely helpful in improving our paper. We tried our best to address all your comments. Thank you again for such detailed comments.

Reviewer 2 Report

The title of the paper should be improved, because word "the River" is not clear. 

The manuscript focuses on flood hazards and sedimentation patterns of the Brahmaputra River, which is of great importance for the risk assessment for the local settlements. Caused by climate change, the magnitude and frequency of floods will increase in future. Furthermore, new embankments along the river have changed the risk and magnitude of floods. This should be also taken into account in the study. The authors gave a very detailed description of the methods and the results. However, in the discussion the impact of embankments should be integrated and discussed in more detail as the maginute of floods is very often influenced by embankments (including the levee effect; for example Ferdous et al. 2019).

The citation style should be unified

L 36 what are such events?
L 88 the °-sign is missing 
L 99 sentence is not clear.
L 104 what is the River
L 126 How can you calculate the population density per length? 
L126 and L130 Please delete the repetition
In the study area the construction of dams and embankments should be also mentioned as they have also an impact on the flood duration (Hofer & Messerli 2006 for Bangladesh, Mustafa et al. 2011 for Indus)
L131 and L131 one link to Figure 1b can be deleted
Figure 1: In the context of the manuscript, international boundaries are not important and should be deleted because of the conflicts between India and Pakistan. I would suggest to enlarge the map to the watershed of the Brahmaputra that more details are visible. As the erosion basis of rivers is the sea level, the bathymetry of the Bay of Bengal is not important and I recommend to use the sea level.

L 175: references are separated

L275 unify the numbers (549 instead of 549.83 )

Figure 5: Unify the figures, either capital or small letters

Figure 6: What is the black line, what is the red box, what the red and blue color?

Tab. 4 needs a new layout; the complete space should be used

L438ff “human settlements are” instead of “human settlement is”

L 488ff I guess that you mean “at the end of the century”
L504 delete the hyphenation (morpho- logical)

Author Response

Thank you for the review, and we appreciate your questions and suggestions.

Reviewer 3 Report

I think this is an interesting manuscript that deals with an interesting case study. The topic of flood hazard under climate change condition is certainly actual and a good fit for the journal. Personally, I am not an expert of flood hazard but of natural hazards in general thus I do not have technical comments on the specifics of the models adopted by the authors. Nevertheless, I think the manuscript can be improved editorially-wise:

- the title is generic and an overshooting of what you actually do. Please restrict it to your case study as you cannot draw general conclusions.

- the structure is ok but the manuscript is too long (which makes it "boring") at times. I suggest that you shorten it, particularly by reducing the number of display items to the essential - this is a research article, not a governmental report. I would only use 1-2 figures for introduction+methods (e.g., study area and a flow chart) + 3-4 essential results/discussion figure. I would put all the rest (including the tables with model parameters) into a supplementary file only to be read by readers who wish to study the case more in detail.

- please also note that some figures are cut: titles on horizontal axes are not visible; also, some figures can clearly be merged, such as 7 and 8

Author Response

Thank you for your positive feedback. All your comments were extremely helpful in improving our paper. We tried our best to address all your comments.

Round 2

Author Response

Thank you for your suggestions. We tried our best to address all of your comments. 
